# Nanoscale coordination polymers induce immunogenic cell death by amplifying radiation therapy mediated oxidative stress

Zhusheng Huang 1,4, Yuxiang Wang1,4, Dan Yao1, Jinhui Wu1,2,3, Yiqiao Hu 1,2,3,5✉ & Ahu Yuan1,2,3,5✉

Radiation therapy can potentially induce immunogenic cell death, thereby priming anti-tumor adaptive immune responses. However, radiation-induced systemic immune responses are very rare and insufficient to meet clinical needs. Here, we demonstrate a synergetic strategy for boosting radiation-induced immunogenic cell death by constructing gadolinium-hemin based nanoscale coordination polymers to simultaneously perform X-ray deposition and glutathione depletion. Subsequently, immunogenic cell death is induced by sensitized radiation to potentiate checkpoint blockade immunotherapies against primary and metastatic tumors. In conclusion, nanoscale coordination polymers-sensitized radiation therapy exhibits biocompatibility and therapeutic efficacy in preclinical cancer models, and has the potential for further application in cancer radio-immunotherapy.

[1] State Key Laboratory of Pharmaceutical Biotechnology, School of Life Science and Medical School, and Jiangsu Key Laboratory for Nano Technology, Nanjing University, Nanjing 210093, China. [2] Jiangsu Key Laboratory for Nano Technology, Nanjing University, Nanjing 210093, China. [3] Institute of Drug R&D, Medical School of Nanjing University, Nanjing 210093, China. [4] These authors contributed equally: Zhusheng Huang, Yuxiang Wang. [5] These authors jointly supervised this work: Yiqiao Hu, Ahu Yuan. ✉email: huyiqiao@nju.edu.cn; yuannju@nju.edu.cn

mmunogenic cell death (ICD) is a specific cell death modality to elicit an immune response against the antigens of dead or dying tumor cells[1,2]. ICD could be induced by chemical (chemotherapies, etc.)[3], physical (ionizing radiation, photodynamic therapeutics, etc.)[4,5] and infective (oncolytic virus, etc.) agents[6], which trigger intracellular stress including reactive oxygen species (ROS) and structural/functional alterations of the endoplasmic reticulum (ER), and the release of damage-associated molecular patterns (DAMPs)[7]. The released or exposed DAMPs could mediate maturation and migration of dendritic cells (DCs), subsequently priming the anti-tumor adaptive immune responses[8]. Unfortunately, in the clinic, many tumors are poorly immunogenic[9], and tumor cells could evade immune surveillance[10–12]. Even after treatment with checkpoint blockade immunotherapies (CBI), including antibodies for cytotoxic T-lymphocyte associated protein 4 (CTLA-4) and programmed-cell-death protein 1 (PD-1), most patients presented insufficient systemic immune responses for tumor regression[13–15].

Radiotherapy (RT) is a widely used tumor treatment that utilizes high-energy ionizing radiation to generate DNA damage in tumor cells[16,17]. Owing to low systemic side effects and definite curative effects, about 50% of tumor patients would receive radiation therapy[18]. Extensive studies have confirmed that RT could activate the immune system by inducing ICD[19,20]. ROS induced by RT could destroy the integrity of the nucleus and release high mobility group protein B1 (HMGB1)[4,8]. Moreover, RT could increase ROS level in the ER, leading to calreticulin (CRT) exposure[21]. These ICD-associated DAMPs could potentially promote the activation and migration of dendritic cells (DCs), which then prime T cells for systemic anti-tumor immune responses[22]. However, radiation-induced systemic immune responses are insufficient to meet clinical needs[23]. According to the literature[24], only 46 cases of radiotherapy-mediated immune activation and subsequent abscopal effects were reported from 1969 to 2014, indicating the huge challenges as well as enormous room for improvement[25–28].

The accumulated evidences suggest that the therapeutic efficacy of radiation is significantly limited by the weak irradiation absorption and endogenous radiation resistance[29]. First, tumor tissues exhibited low X-ray absorption and energy deposition capacities, resulting in inadequate •OH generation[30]. Second, there are vast hypoxic areas within the solid tumors owing to the imbalance of oxygen supply and demand[31–34]. The unusually high concentrations of ROS including hydrogen peroxide ($H_2O_2$) within tumor tissues result in metabolic dysregulation[35,36]. The tumor cells would subsequently develop adaptive antioxidant mechanisms, including catalase, superoxide dismutase, and glutathione (GSH) to maintain the balance of oxidation-reduction (redox)[37,38]. High concentrations of reducing substances such as GSH could rapidly quench •OH generated during RT and weaken therapeutic efficacy[39,40]. All these mechanisms contribute to the insufficient •OH for effectively killing tumor cells and inducing ICD. Therefore, we proposed the possibility to amplify RT mediated oxidative stress to induce ICD for antitumor immunity activation.

In this work, we construct a nanoscale coordination polymers (NCPs) based on gadolinium ($Gd^{3+}$, commonly used in clinic) and 5′-Guanosine monophosphate (5′-GMP, widely existed in the organism) via supramolecular self-assembly, and then integrate Hemin (PANHEMATIN®) with peroxidase-mimic catalytic activity into $Gd^{3+}$/5′-GMP NCPs (Gd-NCPs) to form a novel radiosensitizer Hemin@ $Gd^{3+}$/5′-GMP NCPs (H@Gd-NCPs) (Fig. 1a). Due to the presence of metal element Gd, H@Gd-NCPs can act as a magnetic resonance imaging (MRI) contrast agent, which outperforms Magnevist commonly used in the clinic[41].

Then H@Gd-NCPs effectively enhance X-ray absorption and produce more ROS, especially hydroxyl radical within tumor tissues[42]. In addition, Hemin encapsulated in H@Gd-NCPs can enhance peroxidase-like properties to utilize overexpressed $H_2O_2$ in tumor microenvironment to deplete GSH. The integration of ROS enhancement and GSH depletion eventually amplify irradiation mediated oxidative stress and induce ICD. The antitumor immunity activated by H@Gd-NCPs can further be strengthened by immune checkpoint blockade therapy against primary, distant, and metastatic tumors. In summary, the established H@Gd-NCPs exhibit biocompatibility and powerful ability for radiation sensitization and antitumor immune activation, with great potential for further development (Fig. 1b).

## Results

### Synthesis and characterization of Gd-NCPs and H@Gd-NCPs.
The self-assembly of $GdCl_3$ (10 mM) and 1.5 equimolar 5′-guanosine monophosphate (5′-GMP) in HEPES buffer (pH 7.4) resulted in white precipitates. Then, the precipitates were washed and sonicated to form $Gd^{3+}$ based nanoscale coordination polymers (Gd-NCPs)[43]. Next, $GdCl_3$ (10 mM), 5′-GMP (1.5 equimolar), and Hemin (0.1 equimolar) were mixed and self-assembled in HEPES buffer (pH 7.4), resulting in dark brown precipitates. Then, the precipitates were washed and sonicated to obtain Hemin coordinative Gd-NCPs (H@Gd-NCPs), and Hemin and $Gd^{3+}$ were quantitated, respectively (Supplementary Figs. 1 and 2). Ultra-high-resolution field emission scanning electron microscope (FE-SEM) imaging showed that H@Gd-NCPs presented a nanosphere morphology (Fig. 2a), with an average diameter of 112.5 nm (Dynamic light scattering measurement). Gd-NCPs exhibited a slightly smaller particle size (106.4 nm) than H@Gd-NCPs (Fig. 2b). Gd-NCPs and H@Gd-NCPs exhibited the zeta potential of −5.46 and −5.81 mV, respectively (Fig. 2c).

H@Gd-NCPs and free Hemin showed the same ultraviolet (UV) absorption peak at 390 nm, confirming the presence of Hemin within H@Gd-NCPs (Fig. 2d). Coordination of Hemin within the nanoparticles was further confirmed through Fourier transform infrared spectrum (FT-IR), and H@Gd-NCPs exhibited characteristic absorption peak of $GdCl_3$ (~1620 cm$^{-1}$), 5′-GMP (~1051 and ~1691 cm$^{-1}$) and Hemin (~1620 and ~2922 cm$^{-1}$), respectively (Fig. 2e). The fluorescence spectra of H@Gd-NCPs with various concentrations of Hemin showed a strong fluorescence at 660 nm under the excitation wavelength of 330 nm (Supplementary Fig. 3). The X-ray photoelectron spectroscopy (XPS) also confirmed the presence of metal element Gd with characteristic binding energy at 148.00 eV (Gd $4d_{3/2}$) and Fe with characteristic binding energy at 711.75 eV (Fe $2p_{3/2}$), consistent with standard XPS spectrum of $Gd^{3+}$ and $Fe^{3+}$ (NIST XPS Database), respectively. Other non-metallic elements such as C, N, O, P, Cl were also detected in H@Gd-NCPs (Fig. 2f and Supplementary Fig. 4). Furthermore, the stability of the Gd-NCPs and H@Gd-NCPs were measured by DLS at 25 °C and 37 °C (dispersed in saline or 50% bovine serum), and the results showed that Gd-NCPs and H@Gd-NCPs were stable within 48 h (Fig. 2g).

We next performed the dialysis experiments of Gd-NCPs and H@Gd-NCPs to evaluate their stability. Gd-NCPs and H@Gd-NCPs were packed into dialysis bags, followed by dialysis in 50% bovine serum solution or deionized water for 7 days, respectively. The dialysates were concentrated to detect free $Gd^{3+}$ by colorimetry[43]. As shown in Supplementary Table 1, almost no free $Gd^{3+}$ could be detected in the dialysates after 7 days' dialysis. These results suggested that the Gd-NCPs and H@Gd-NCPs could maintain stable in deionized water or serum. To further evaluate whether the H@Gd-NCPs would undergo

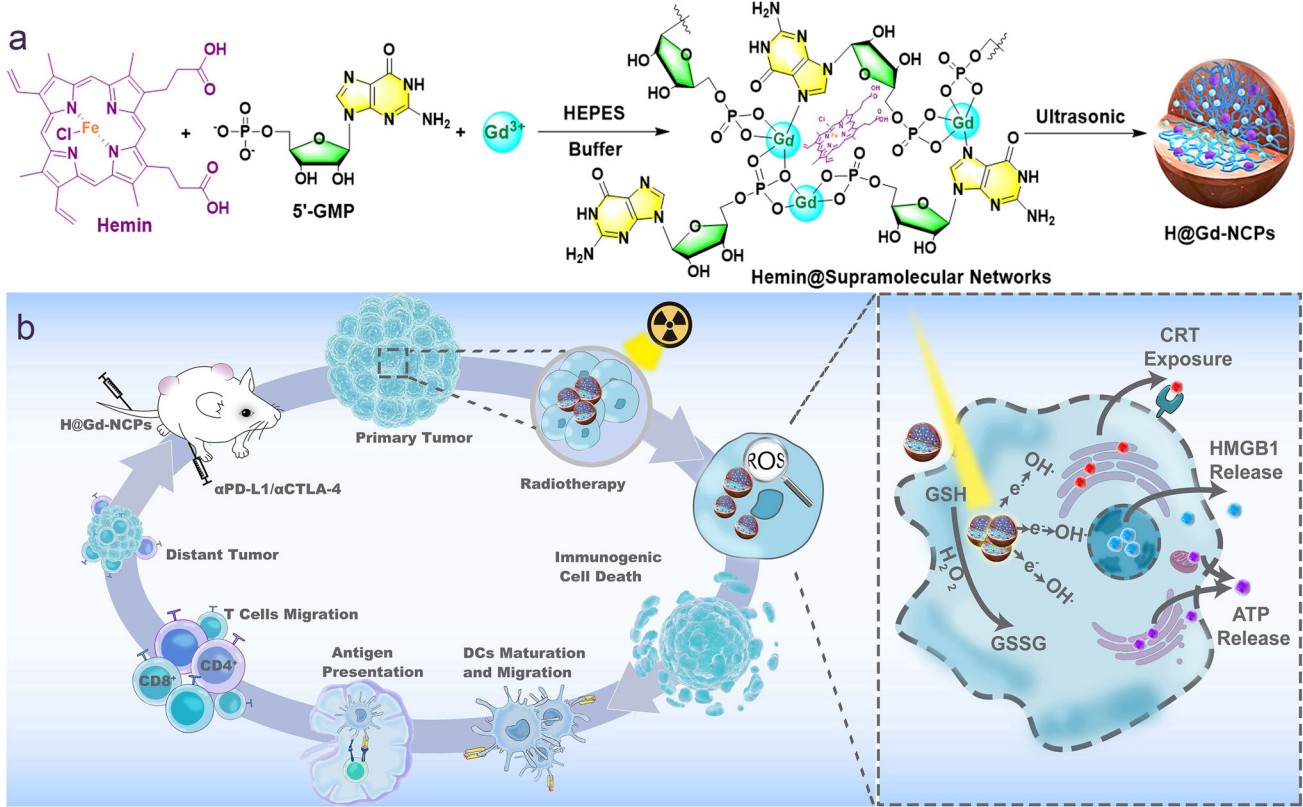

**Fig. 1 The preparation and mechanism of H@Gd-NCPs. a** Schematic illustration of the preparation of nanoscale coordination polymers H@Gd-NCPs. **b** The mechanism of H@Gd-NCPs for radiosensitization via amplifying intracellular oxidative stress to potentiate checkpoint blockade immunotherapies. Dendritic cells (DCs), glutathione (GSH), oxidized glutathione (GSSG), hydroxyl radicals (•OH), calreticulin (CRT), high mobility group protein B1 (HMGB1), adenosine triphosphate (ATP).

trans-metallation in physiological conditions, H@Gd-NCPs were packed into dialysis bags, stirred in 100.0 mL dialysates (50% bovine serum, adding extra $[Na^+] = 150$ mM, $[K^+] = 5.0$ mM, $[Ca^{2+}] = 2.5$ mM, $[Mg^{2+}] = 1.25$ mM, $[Zn^{2+}] = 30$ μM, $[Fe^{3+}] = 30$ μM, $[Cu^{2+}] = 30$ μM to mimic physiological environment) for 7 days at pH = 7.4, 6.5, and 5.0, respectively. The dialysates were collected and concentrated by vacuum distillation, and then the concentrates were analyzed by ICP-OES (Avio 500, USA). As shown in Supplementary Fig. 5, all the above metal ions, except $Gd^{3+}$, could be detected in dialysates at various pH, potentially indicating that obvious transmetalation process could not be proven under these applied conditions.

**Hydroxyl radical formation and GSH elimination in vitro.** Hydroxyl radicals (•OH) were the main cytotoxic radical species from ionizing radiation and could be detected via the methylene blue (MB) bleaching method[44]. We compared the •OH yield of the $H_2O$, $GdCl_3$, Gd-NCPs, and H@Gd-NCPs upon various radiation doses (Fig. 2h). The incorporation of high atomic number (High-Z) element gadolinium ($GdCl_3$, Gd-NCPs, and H@Gd-NCPs) could accelerate the decay of MB, indicating that Gd in a free state or nanoparticles could enhance X-ray absorption and energy deposition to promote •OH generation. Hemin could mimic peroxidase to decompose $H_2O_2$ by utilizing GSH as an electron donor, which in turn was oxidized into GSSG (Supplementary Fig. 6)[45–47]. After encapsulation, H@Gd-NCPs enhanced the peroxidase-like activity of Hemin by avoiding their own π-π stack quenching (Fig. 2i). These results confirmed the dual sensitization mechanisms of H@Gd-NCPs on RT to achieve antitumor therapeutic effects.

**Cellular uptake and intracellular behavior of H@Gd-NCPs in vitro.** Next, we detected the therapeutic effects of H@Gd-NCPs on tumor cells. First, CT26 colorectal tumor cells were used to detect their intracellular uptake behavior, and DAPI (blue fluorescence) and Lysotracker (green fluorescence) were employed to mark the nucleus and lysosomes, respectively. As shown in Fig. 3a, tumor cells treated with H@Gd-NCPs showed red fluorescence by confocal laser scanning microscope (CLSM) due to the presence of Hemin, which was colocalized with green fluorescence of Lysotracker. These results showed that H@Gd-NCPs could be endocytosed within tumor cells and mainly located in lysosomes. In addition, the 2′,7′-dichlorodihydrofluorescein diacetate ($H_2DCFDA$) was employed as a probe to monitor intracellular ROS generation induced by Gd-NCPs or H@Gd-NCPs. Upon radiation, CT26 colorectal tumor cells incubated with H@Gd-NCPs showed stronger intracellular green fluorescence than Gd-NCPs or PBS. Without radiation, there was no ROS generation in either PBS, Gd-NCPs or H@Gd-NCPs groups (Fig. 3b, c). These results showed that H@Gd-NCPs could dramatically decrease the intracellular GSH/GSSG ratio in CT26 colorectal tumor cells, but Gd-NCPs did not (Fig. 3d).

We subsequently detected the cytotoxicity of various concentrations of Gd-NCPs and H@Gd-NCPs on CT26 tumor cells with or without irradiation. At 200 μM $[Gd^{3+}]$, both Gd-NCPs and H@Gd-NCPs did not exhibit obvious cytotoxicity to CT26 cells, indicating their biocompatibility (Fig. 3e). Upon radiation, Gd-NCPs inhibited tumor cells better than radiation alone in a dose-dependent manner, indicating that Gd mediated X-ray deposition could induce more cytotoxic hydroxyl radical to kill tumor cells. Furthermore, H@Gd-NCPs inhibited tumor cells better than Gd-NCPs, which should be attributed to GSH elimination (Fig. 3f).

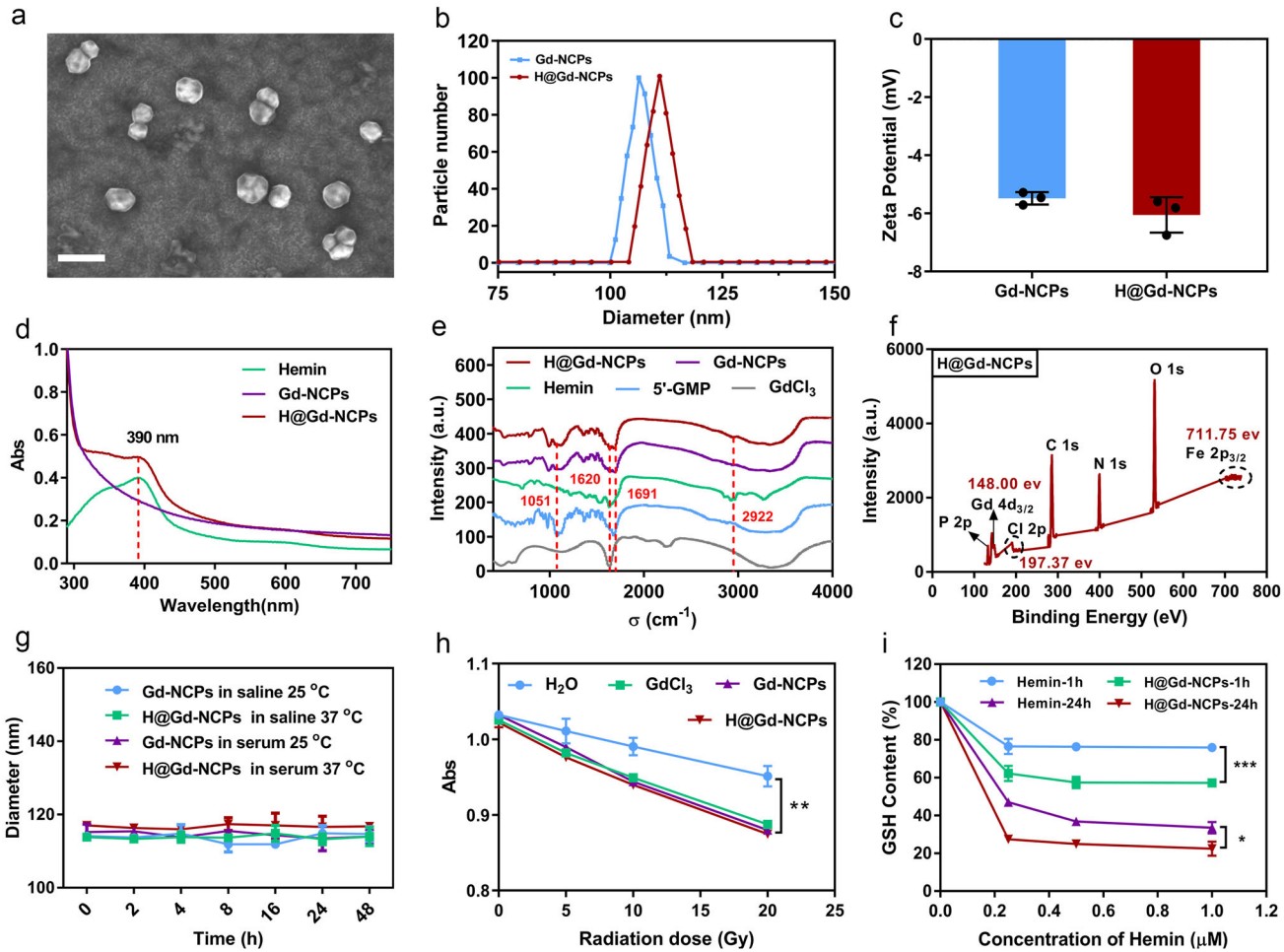

**Fig. 2 Characterization of Gd-NCPs and H@Gd-NCPs. a** Ultra-high-resolution field emission scanning electron microscope (FE-SEM) imaging of H@Gd-NCPs, scale bar = 200 nm. **b** Particle size of H@Gd-NCPs and Gd-NCPs measured by dynamic light scattering ($n = 3$ biologically independent samples). **c** Zeta potential of Gd-NCPs and H@Gd-NCPs ($n = 3$ biologically independent samples). **d** Normalized UV-vis spectra of Hemin, Gd-NCPs and H@Gd-NCPs. **e** Fourier transform infrared (FT-IR) spectrum of 5′-GMP, $GdCl_3$, Hemin, Gd-NCPs, and H@Gd-NCPs. **f** Qualitative element analysis of H@Gd-NCPs by X-ray photoelectron spectroscopy (XPS). **g** Dynamic light scattering data of Gd-NCPs and H@Gd-NCPs incubated with saline or 50% serum at 25 or 37 °C, respectively ($n = 3$ biologically independent samples). **h** Comparison of reactive oxygen species (ROS) production between $H_2O$, $GdCl_3$, Gd-NCPs and H@Gd-NCPs groups ([$Gd^{3+}$] = 20 μM) under various radiation doses as determined by the decay of methylene blue absorption (Abs) at $\lambda = 664$ nm ($n = 3$ biologically independent samples, **$p = 0.0049$). **i** Concentration and time-dependent glutathione (GSH) elimination by free Hemin and H@Gd-NCPs in vitro ($n = 3$ biologically independent samples, ***$p = 0.0001$, *$p = 0.0463$). All experiments were repeated twice independently with similar results. All data were presented as mean ± SD. Two-sided Student's $t$-test was used to calculate the statistical difference between two groups. *$p < 0.05$, **$p < 0.01$, ***$p < 0.001$. Source data are provided as a Source data file.

To further investigate the long-term radiation-sensitization of H@Gd-NCPs, the cell cloning assay was performed. As shown in Supplementary Fig. 7, there were only a few viable cell colonies (12 clones) in the H@Gd-NCPs+RT group. While in Saline, Saline+RT, and Gd-NCPs + RT groups, the tumor cell colonies were 575, 209, and 123, respectively. These results indicated that H@Gd-NCPs could effectively sensitize radiation to prevent tumor cell proliferation. All these results demonstrated that the biocompatible radiosensitizer H@Gd-NCPs could enhance intracellular oxidative stress for superior therapeutic efficiency.

**Accumulation and MRI of H@Gd-NCPs in tumor tissues.** The chelate of paramagnetic $Gd^{3+}$ has been widely used as MRI contrast agent in clinical practice due to the acceleration of $T_1$ relaxation (Fig. 4a)[48]. Thus we evaluated the performance of Magnevist (commercial Gd contrast agent), Gd-NCPs, and H@Gd-NCPs in vitro. As shown in Fig. 4b, $T_1$-weighted images

from Gd-NCPs and H@Gd-NCPs were superior to those from commercial MRI contrast agent Magnevist. The longitudinal relaxivities ($r_1$) values of Magnevist, Gd-NCPs, and H@Gd-NCPs were 3.422, 4.028, and 3.821 mM$^{-1}$ s$^{-1}$ under the magnetic field of 7.0 Tesla (T), respectively, which indicated that the coordination and encapsulation did not influence MRI contrast ability of $Gd^{3+}$ (Fig. 4c). As shown in Fig. 4d, e, two hours after intravenous injection of Magnevist, the tumor region exhibited a higher signal than surrounding tissues, and the MR signal intensity of Magnevist decayed gradually in the following 6~60 h. The MRI signal of H@Gd-NCPs reached maximum at 6 h post-injection in the tumor region and maintained up to 24 h (Fig. 4f). H@Gd-NCPs showed better MRI performance than Magnevist, due to their enhanced accumulation within tumor tissues. Magnevist and H@Gd-NCPs were metabolized through the kidneys, but Magnevist exhibited a rapid renal clearance (Fig. 4g). Meanwhile, the livers in Magnevist and H@Gd-NCPs groups did not show an obvious change in MRI signal intensity (Fig. 4h).

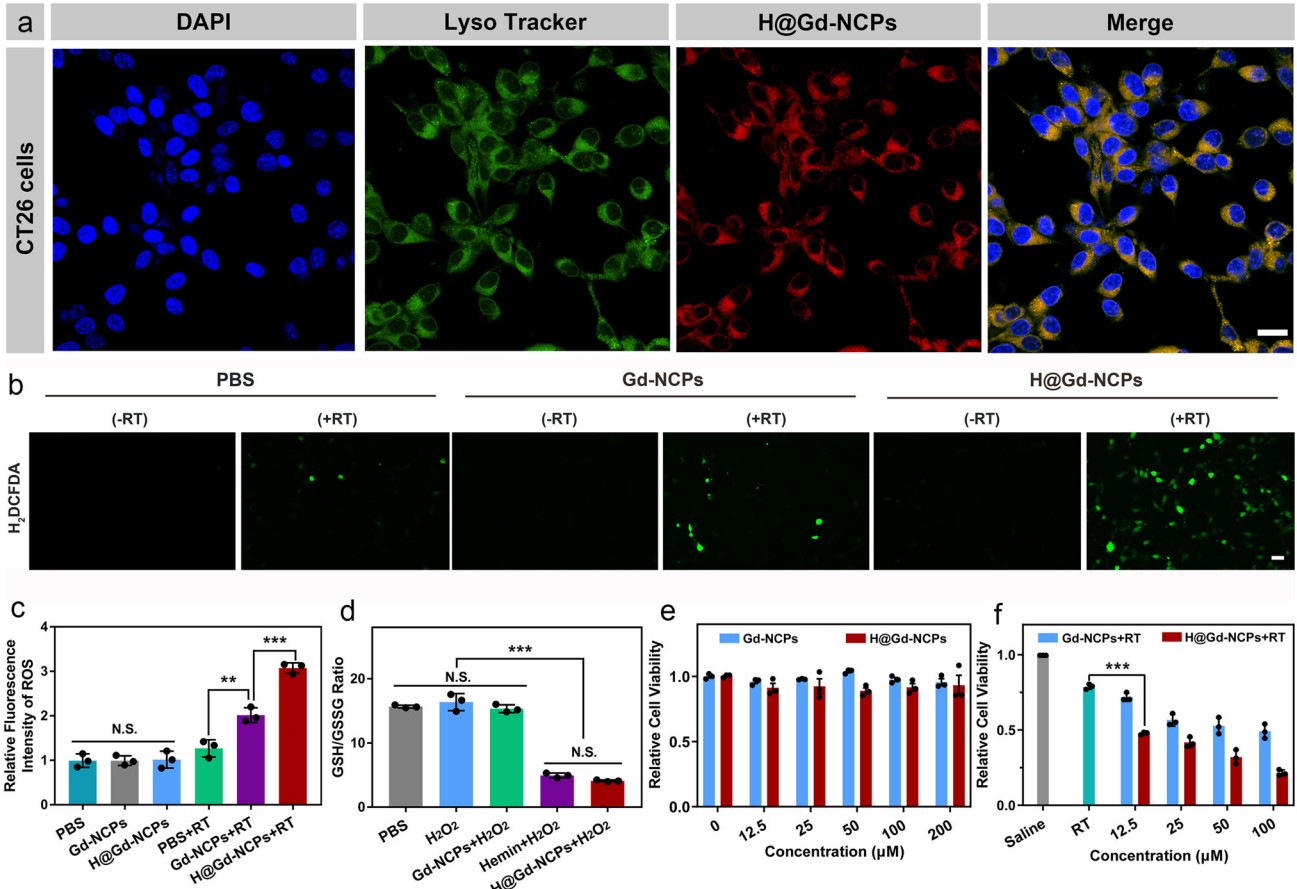

**Fig. 3 Cellular uptake and radiosensitization of H@Gd-NCPs in vitro. a** Confocal laser scanning microscope (CLSM) images of CT26 cells after treatment with DAPI, Lysotracker and H@Gd-NCPs, respectively. Yellow regions indicated localization of H@Gd-NCPs in the lysosomes, scale bar = 20 μm. **b** Fluorescence images of CT26 cells treated with PBS, Gd-NCPs, and H@Gd-NCPs with or without 8 Gy × 1 irradiation and detected with a reactive oxygen species (ROS) probe $H_2DCFDA$ (green fluorescence) for intracellular ROS evaluation, scale bar = 200 μm. **c** Quantification of fluorescence based on (**b**) by ImageJ software ($n = 3$ biologically independent cells, $**p = 0.0071$, $***p = 0.0008$). **d** PBS, Hemin, Gd-NCPs and H@Gd-NCPs decrease intracellular GSH/GSSG ratios in CT26 cells ($n = 3$ biologically independent cells, $***p = 0.0001$). **e** The cytotoxicity of Gd-NCPs and H@Gd-NCPs without irradiation ($[Gd^{3+}] = 0, 12.5, 25, 50, 100, 200$ μM) ($n = 3$ biologically independent cells). **f** The cytotoxicity of Gd-NCPs and H@Gd-NCPs against CT26 cells with 8 Gy × 1 irradiation ($[Gd^{3+}] = 0, 12.5, 25, 50, 100$ μM) ($n = 3$ biologically independent cells, $***p = 0.0002$). All experiments were repeated twice independently with similar results. All data were presented as mean ± SD. Two-sided Student's $t$-test was used to calculate the statistical difference between two groups. N.S. represented non-significance, and $**p < 0.01$, $***p < 0.001$. Source data are provided as a Source data file.

These results demonstrated that H@Gd-NCPs could effectively accumulate in tumor tissues for in vivo MR imaging and radiation sensitization.

MRI signal could qualitatively determine whether there were nanomedicines in tumor tissues, but could not quantify the drug concentration accumulated within tumor tissues. To further verify the distribution of H@Gd-NCPs in the tumors, we also detected their accumulation via a colorimetric method[43]. As shown in Supplementary Fig. 8, after intravenous injection of H@Gd-NCPs, tumor tissues were respectively collected from the CT26 tumor-bearing mice at 2, 6, 12, 24, 48, and 60 h after mice sacrificed. Thymolphthalein complexone (TC) was used as a colorimetric reagent to detect gadolinium in a free state, but not in coordination state. We first tested the tumor tissues extracts without burning and nitrification, and almost no free $Gd^{3+}$ could be detected in H@Gd-NCPs and Magnevist groups (Supplementary Fig. 8a, b). While after burning and nitrification, gadolinium accumulated within tumor tissues in both groups could be detected, respectively. The concentration of H@Gd-NCPs in the tumor tissues peaked at 6 h (1.04 μmol g$^{-1}$ tumor tissue) post-injection and

maintained up to 24 h (0.77 μmol g$^{-1}$ tumor tissue). While the concentration of Magnevist peaked at 2 h (0.74 μmol g$^{-1}$ tumor tissue) post-injection in the tumor regions and exhibited rapid metabolization (Supplementary Fig. 8c–e). These results indicated that the H@Gd-NCPs was accumulated in the tumor tissues, which were similar to MRI results.

Next, we tried to reveal the metabolism process of these nanomedicines via a simulation method. Specifically, we used bovine serum albumin solution (50 mg/mL, 37 °C) as the simulated plasma to continuously dilute H@Gd-NCPs. With the process of dilution, we found that the particle size of H@Gd-NCPs was gradually decreasing from about 100 nm to 5~10 nm (512-fold dilution, Supplementary Fig. 9a–h). These smaller nanoparticles could potentially be metabolized trough the kidneys. Based on this hypothesis, we further detected the state of the metabolic products in the urine of treated mice. We collected urine from mice at 24–48 h after intravenous injection of H@Gd-NCPs. Similarly, we should be able to directly detect free $Gd^{3+}$ via the colorimetric method if there was free $Gd^{3+}$ in the urine. However, the results of direct testing indicated that there was almost no free $Gd^{3+}$ in urine (Supplementary Fig. 10a).

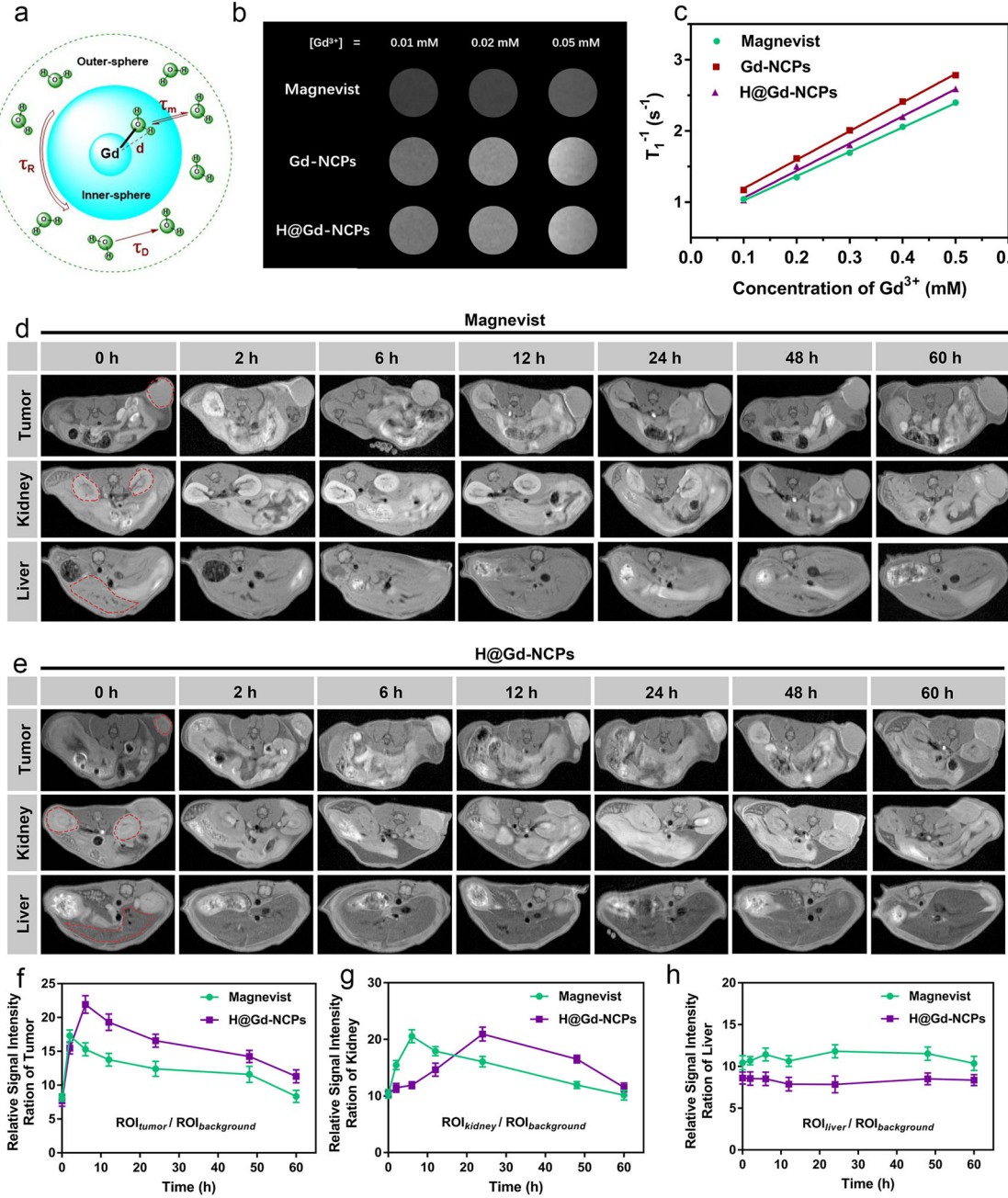

**Fig. 4 Magnetic resonance imaging (MRI) in vitro and in vivo. a** Schematic diagram of MRI. **b** $T_1$-weighted MR images of Magnevist, Gd-NCPs, and H@Gd-NCPs at pH 7.4 ($n = 3$ biologically independent samples), this experiment was repeated twice independently with similar results. **c** Determination of longitudinal relaxivities ($r_1$) values for Magnevist, Gd-NCPs and H@Gd-NCPs ($n = 3$ biologically independent samples), this experiment was repeated twice independently with similar results. **d** Dynamic MR imaging after intravenous injection of Magnevist ([Gd³⁺] = 30 mg kg⁻¹), and the dashed red circles indicated tumor, kidney, or liver ($n = 3$ biologically independent animals). **e** Dynamic MRI after intravenous injection of H@Gd-NCPs ([Gd³⁺] = 30 mg kg⁻¹) in vivo, and the dashed red circles indicated tumor, kidney and liver, respectively ($n = 3$ biologically independent animals). **f–h** Relative background signal intensity ration of tumor (**f**), kidney (**g**) and liver (**h**) regions based on Magnevist MR images (**d**) and H@Gd-NCPs MR images (**e**) at different time points ($n = 3$ biologically independent samples). All data were presented as mean ± SD. Source data are provided as a Source data file.

We then burned and nitrified the urine sample, which showed that there was detectable gadolinium (Supplementary Fig. 10b). Therefore, it was reasonable to assume that H@Gd-NCPs became smaller (5~10 nm) through the serial dilution process after intravenous injection. Then, these smaller nanoparticles could be gradually metabolized through the kidneys in the coordination state rather than in a free state. All these results indicated that H@Gd-NCPs could maintain the coordination state during blood circulation and even after renal excretion.

**Enhanced therapeutic efficacy on CT26 tumor in vivo.** We subsequently evaluated the in vivo antitumor efficacy of Gd-NCPs and H@Gd-NCPs in CT26-tumor-bearing *Balb/c* mice. When the tumors reached 80–100 mm³, all mice were randomly divided into six groups, including Saline, Gd-NCPs and H@Gd-NCPs groups with or without RT. Saline, Gd-NCPs ([Gd³⁺] = 30 mg kg⁻¹) or H@Gd-NCPs ([Gd³⁺] = 30 mg kg⁻¹ and [Hemin] = 12.5 mg kg⁻¹) was intravenously injected into the mice, followed by X-ray irradiation (0 or 6 Gy × 2 with fractions

delivered 6 days apart) 6 h post injection. Drug administration and X-ray irradiation were performed on day 0 and 6, respectively. The mice in Saline, Gd-NCPs and H@Gd-NCPs groups were sacrificed when tumor volumes reached 2000 mm$^3$ (day 14), and the mice in the other three groups were sacrificed on day 21 after tumor treatment. Then the tumors were excised and photographed (Supplementary Fig. 11).

As shown in Fig. 5a, Gd-NCPs and H@Gd-NCPs groups without irradiation showed almost no tumor growth inhibition compared to Saline group on day 14. Upon irradiation, Gd-NCPs exhibited radiosensitization effects and caused significant tumor regression. In addition, H@Gd-NCPs effectively eliminated GSH within tumor tissues, enhanced intracellular oxidative stress, and showed the highest tumor inhibition ratio in all groups (Supplementary Fig. 12). The tumor growth inhibition in CT26 colorectal model was confirmed by the weights of excised tumors on day 14 (without irradiation) or day 21 (with irradiation) (Fig. 5b). We found no significant difference in body weight among RT, Gd-NCPs + RT and H@Gd-NCPs + RT groups, indicating the bio-safety of H@Gd-NCPs during treatments (Fig. 5c and Supplementary Fig. 13). Serum biochemistry analysis and histological analysis (H&E) of major organs showed no significant difference in all groups, further confirming the safety of H@Gd-NCPs (Supplementary Figs. 14 and 15). Immunohistochemical (IHC) staining of Ki67 showed that the highly proliferative tumor cells were much less after H@Gd-NCPs + RT treatment compared with other five groups (Fig. 5d, e). TUNEL staining indicated more apoptotic tumor cells in H@Gd-NCPs +RT group than in RT or Gd-NCPs + RT groups (Fig. 5d, f). These results suggested that the combination of High-Z effect and GSH elimination could significantly amplify intracellular oxidative stress for tumor cell inhibition. The formation of γ-H2Aχ is a key marker of double-strand DNA breaks after X-ray irradiation. As expected, three groups without irradiation including Saline, Gd-NCPs, and H@Gd-NCPs exhibited little scattered green fluorescence, but H@Gd-NCPs + RT induced most double-strand DNA breaks in all groups, demonstrating their radiosensitization effects (Fig. 5d, g). H&E staining of tumor sections confirmed the therapeutic efficacy of H@Gd-NCPs+RT, which caused the largest tumor necrosis regions (Fig. 5d). Therefore, H@Gd-NCPs mediated oxidative stress amplification could inhibit tumor cell proliferation and tumor growth.

**Induction of ICD**. ICD induction was evaluated by CRT exposure, HMGB1 release and adenosine triphosphate (ATP) secretion[1–8]. As shown in Fig. 6a, CT26 cells treated with Saline, RT (8 Gy × 1), and Gd-NCPs+RT (8 Gy × 1) showed fewer cell-surface exposure of CRT, but tumor cells treated by H@Gd-NCPs + RT (8 Gy × 1) presented much more CRT and quantitative analysis demonstrated the superiority of H@Gd-NCPs in inducing ICD (Fig. 6b). Compared with Saline, RT, and Gd-NCPs + RT groups, H@Gd-NCPs+RT significantly enhanced the release of HMGB1 and ATP secretion from CT26 cells (Fig. 6c, d). In addition, western blot analysis of CT26 tumor tissues showed that the extracellular and cytoplasmic HMGB1 was significantly enhanced in H@Gd-NCPs+RT (6 Gy × 1) group compared with Saline, RT (6 Gy × 1), Gd-NCPs + RT (6 Gy × 1) groups (Fig. 6e). These results indicated that H@Gd-NCPs had the ability to effectively amplify oxidative stress, leading to the exposure or release of DAMPs and ICD. To further investigate the influence of DAMPs exposure or release on DC maturation, CT26 colorectal tumor-draining lymph nodes were collected for flow cytometry analysis. As shown in Fig. 6f, CT26-bearing mice treated with Gd-NCPs or H@Gd-NCPs did not show change in the expression of CD80$^+$ CD86$^+$ in CD11c$^+$ DCs. After radiation therapy (6 Gy × 1),

H@Gd-NCPs significantly enhanced the ratio (57.85%) of mature DCs (CD11c$^c$ CD80$^+$ CD86$^+$), higher than that in radiation (37.42%) or Gd-NCPs + RT (43.45%) groups. All these results indicated that amplification of oxidative stress could induce potent immunogenicity and DC maturation, which would potentially activate systemic anticancer immunity against distant or metastatic tumors.

**Abscopal effect of H@Gd-NCPs-sensitized radiation therapy**. As H@Gd-NCPs-sensitized radiation therapy is highly immunogenic, we then evaluated the synergistic effects with immune checkpoint blockade (anti-PD-L1 antibody, αPD-L1) for systemic antitumor therapy. As shown in Fig. 7a, H@Gd-NCPs+RT treatment successfully inhibited the primary tumors with or without αPD-L1 in the initial period. However, tumors treated by H@Gd-NCPs + RT grew slowly in the late therapeutic period, while the tumors remained to be suppressed after combination with αPD-L1. Notably, H@Gd-NCPs + RT exhibited abscopal effect, while radiation therapy alone did not, which should be ascribed to adequate ICD induction and the activation of antitumor immunity. After synergism with αPD-L1, 37.5% mice were tumor free at day 21, indicating complete regression of irradiated and non-irradiated tumors (Fig. 7b). These results showed that H@Gd-NCPs sensitized irradiation synergized with the immune checkpoint blockade to effectively stimulate systemic antitumor immune response. All mice were sacrificed on day 21 after tumor treatments, the weight of excised tumor tissues (Fig. 7c, d) and the growth curves of individual tumor (Supplementary Figs. 16 and 17) further confirmed the synergistic therapeutic efficacy.

**Activation of systemic antitumor immunity**. Next, we investigated the mechanism of systemic antitumor immunity induced by H@Gd-NCPs. We profiled infiltrating leukocytes, including CD4$^+$ T cells (CD3$^+$ CD4$^+$ T cells) and CD8$^+$ T cells (cytotoxic T lymphocytes, CTLs, CD3$^+$ CD8$^+$ T cells) in primary and distant tumors. CD4$^+$ or CD8$^+$ T cell infiltration did not change after irradiation alone, which indicated that commonly used dose of X-ray irradiation was insufficient to alter the immunological tumor microenvironment[23,24]. However, H@Gd-NCPs sensitized irradiation enhanced the CD4$^+$/CD8$^+$ T cell infiltration (4.03%/ 1.29% in primary tumors, 3.99%/1.63% in distant tumors), compared with radiation alone (1.55%/0.56% in primary tumors, 1.05%/0.69% in distant tumors). In synergy with αPD-L1, H@Gd-NCPs+RT further enhanced the ratio of infiltrated CD4$^+$/CD8$^+$ T cells to 5.95%/1.94% in primary tumors and 5.98%/2.31% in distant tumors. Notably, tumor-infiltrating CD4$^+$/CD8$^+$ T cells were significantly more than those in RT + αPD-L1 group (2.65%/0.91% in primary tumors, 2.64%/1.04% in distant tumors), indicating that H@Gd-NCPs mediated radio-sensitization created a favorable immunological microenvironment for antitumor therapy (Fig. 7e–h). Moreover, immunofluorescence (Supplementary Fig. 18) and immuno-histochemistry (IHC) (Supplementary Figs. 19 and 20) images of tumor-infiltrating CD4$^+$ T cells and CD8$^+$ T cells confirmed the results of flow cytometry.

Cytotoxic T lymphocytes could release interferon-γ (IFN-γ) to promote the immune microenvironment[12]. Thus we detected IFN-γ in both primary and distant tumors by ELISA. The secretion of IFN-γ was significantly enhanced in H@Gd-NCPs + RT group, and could be further enhanced by synergism with αPD-L1 to achieve optimal tumor immunotherapeutic efficacy (Fig. 7I, j). In addition, the steady plot of body weights in all treated mice demonstrated the safety of H@Gd-NCPs + RT in vivo (Fig. 7k and Supplementary Fig. 21). To further confirm the immunological memory responses induced by H@Gd-NCPs

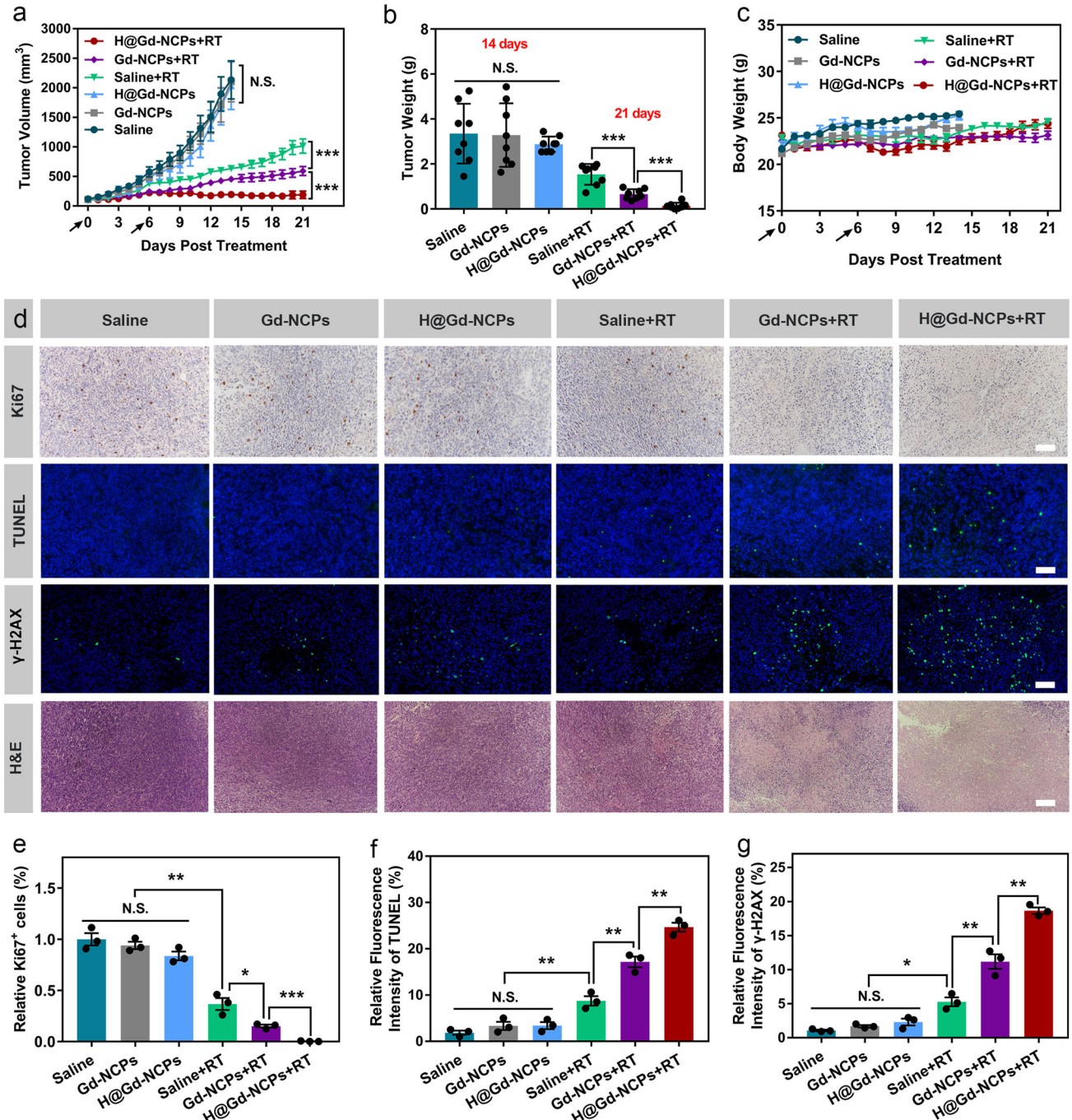

**Fig. 5 Therapeutic efficacy of H@Gd-NCPs in CT26-bearing mice. a** Tumor growth curves after various treatments ([Gd$^{3+}$] = 30 mg kg$^{-1}$ and [Hemin] = 12.5 mg kg$^{-1}$) with or without irradiation. Treatments were performed on days 0 and 6. X-ray radiation therapy was performed 6 h after nanomedicines intravenous injection (black arrow). RT 6 Gy × 2 with fractions delivered 6 days apart (n = 8 biologically independent animals, ***p = 0.0001). Data were presented as mean ± SEM. **b** Tumor weight without irradiation groups collected on day 14, tumor weight with irradiation groups collected on day 21 (n = 8 biologically independent animals, ***p = 0.0001). **c** Dynamic body weight of CT26-bearing mice in different groups during treatments (n = 8 biologically independent animals). **d** Images of Ki67 immunohistochemical staining of tumor slices, immunofluorescence images of tumor slices stained with TUNEL assay kit and γ-H2Aχ antibody and H&E sections, scale bar = 100 μm. The γ-H2Aχ tumor slices were harvested 24 h after radiotherapy (6 Gy ×1) and the Ki67, TUNEL tumor slices were harvested 48 h after radiotherapy (6 Gy ×1). These experiments were repeated twice independently with similar results. **e–g** Quantification of the relative percentage of (**e**) Ki67 positive cells (**p = 0.0016, *p = 0.0227, ***p = 0.0005) (**f**) TUNEL (**p = 0.0035, **p = 0.0053, **p = 0.0072) and γ-H2Aχ (*p = 0.0364, **p = 0.0093, **p = 0.0031) mean fluorescence intensity after different treatments (n = 3 biologically independent animals). Data (**b**, **c**, **e–g**) were presented as mean ± SD. Two-sided Student's t-test was used to calculate the statistical difference between two groups. N.S. represented non-significance, and *p < 0.05, **p < 0.01, ***p < 0.001. Source data are provided as a Source data file.

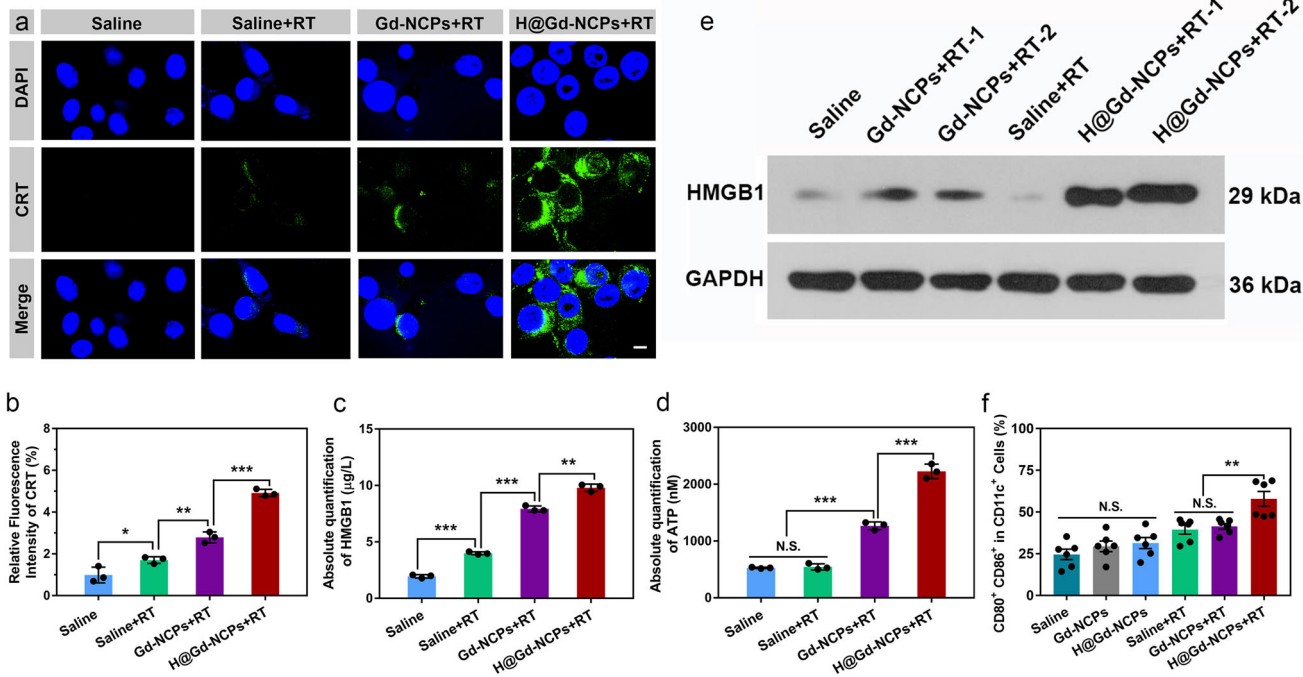

**Fig. 6 Immunogenic cell death induction in vitro and in vivo. a** Immunofluorescence of CT26 colorectal tumor cells stained with calreticulin (CRT) antibody ($n = 3$ biologically independent cells), RT 0 or 8 Gy × 1, scale bar = 5 μm. This experiment was repeated twice independently with similar results. **b** Quantification of relative CRT means fluorescence intensity after different treatments ($n = 3$ biologically independent cells, $*p = 0.0389$, $**p = 0.0037$, $***p = 0.0003$). **c** Detection of high mobility group protein B1 (HMGB1) release by ELISA kit ($n = 3$ biologically independent cells, $***p = 0.0001$, $***p = 0.0001$, $**p = 0.0014$), RT 0 or 8 Gy × 1. This experiment was repeated twice independently with similar results. **d** Detection of adenosine triphosphate (ATP) secretion by luciferin-based ATP assay kit ($n = 3$ biologically independent cells, $***p = 0.0001$, $***p = 0.0003$), RT 0 or 8 Gy × 1. This experiment was repeated twice independently with similar results. **e** Western blot of HMGB1 in CT26-bearing mice tumor tissues after various treatments, the tumor tissues were harvested 48 h after radiotherapy (0 or 6 Gy × 1, $n = 3$ biologically independent animals). This experiment was repeated once independently with similar results. **f** Flow cytometry analysis of dendritic cells (DCs) maturation in tumor-draining lymph nodes (TDLNs), the TDLNs were harvested 5 days after radiotherapy (0 or 6 Gy × 1, n = 6 biologically independent animals, $**p = 0.0063$). All data were presented as mean ± SD. Two-sided Student's $t$-test was used to calculate the statistical difference between two groups. N.S. represented non-significance, and $*p < 0.05$, $**p < 0.01$, $***p < 0.001$. Source data are provided as a Source data file.

+ RT, the effector memory T cells (T$_{EM}$, CD3$^+$ CD8$^+$ CD44$^+$ CD62L$^-$) and central memory T cells (T$_{CM}$, CD3$^+$ CD8$^+$ CD44$^+$ CD62L$^+$) in the excised spleens in various groups were analyzed by flow cytometry. Different from T$_{CM}$ cells, T$_{EM}$ cells could directly induce potent immunological memory protection through secreting antitumor cytokines such as TNF-α and IFN-γ[49–51]. The percentage of T$_{EM}$ cells after H@Gd-NCPs + RT + αPD-L1 treatment was much higher than that in mice treated by RT + αPD-L1 or H@Gd-NCPs + RT (Fig. 7l). These results indicated that H@Gd-NCPs sensitized radiation could induce long-term immunological memory to synergize with CBI to potentially eradicate residual tumors.

**CD8$^+$ T cells depletion experiments and ex vivo analysis of immune cells.** We attempted to clarify whether CD8$^+$ T cells play an important role in the anti-tumor effects by H@Gd-NCPs sensitized radiation. Then we performed the CD8$^+$ T cells depletion experiment on a bilateral model of CT26 tumors. As shown in Fig. 8 and Supplementary Fig. 22, we observed that H@Gd-NCPs + RT treatment lost most of the immunotherapeutic effects in primary CT26 tumors after CD8$^+$ T cells depletion (Fig. 8a, c, e). Furthermore, in secondary tumors, CD8$^+$ T cells depletion completely eliminated the therapeutic effect of H@Gd-NCPs + RT treatment (Fig. 8b, d, f). We then analyzed infiltrating cytotoxic CD8$^+$ T cells in primary and distant tumors, respectively. H@Gd-NCPs sensitized irradiation remained the effective CD8$^+$ T cell infiltration (1.26% in primary tumors and

1.68% in distant tumors), when compared with control or RT alone. Subsequently, αCD8a treatment also significantly eliminated H@Gd-NCPs + RT mediated CD8$^+$ T cell infiltration in primary (0.21%) and distant (0.30%) tumors, respectively (Fig. 8g, h). These results indicated that CD8$^+$ T cells deeply involved in H@Gd-NCPs mediated radiation sensitization and immunotherapeutics.

We further detected the ratios of tumor-associated macrophages (TAMs) within the tumor tissues after various treatments (Control, RT, and H@Gd-NCPs + RT) to preliminarily illustrate their roles in H@Gd-NCPs mediated anti-tumor therapy. The ratios of TAMs in whole tumor tissues of RT and H@Gd-NCPs + RT groups did not exhibit obvious change, when compared with control group (Fig. 8i and Supplementary Fig. 23, Supplementary Table 2). We next co-cultured tumor cells and macrophages, dosed with H@Gd-NCPs for 6 h, and then labeled macrophages with PE-F4/ 80-antibody. As shown in Supplementary Fig. 24, CT26 tumor cells exhibited obviously stronger red punctate fluorescence signals, potentially indicating their higher internalization efficiency than macrophages (RAW264.7 cells). We then performed the in vitro cytotoxicity study upon tumor cells and macrophages, respectively. Without radiation, H@Gd-NCPs (0~100 μM of Gd$^{3+}$) did not exhibit obvious cytotoxicity to both CT26 tumor cells and RAW264.7 cells, potentially indicating their great biocompatibility. Upon radiation, H@Gd-NCPs showed superior proliferation inhibition in CT26 tumor cells than RAW264.7 cells, which should be probably attributed to their higher cellular internalization

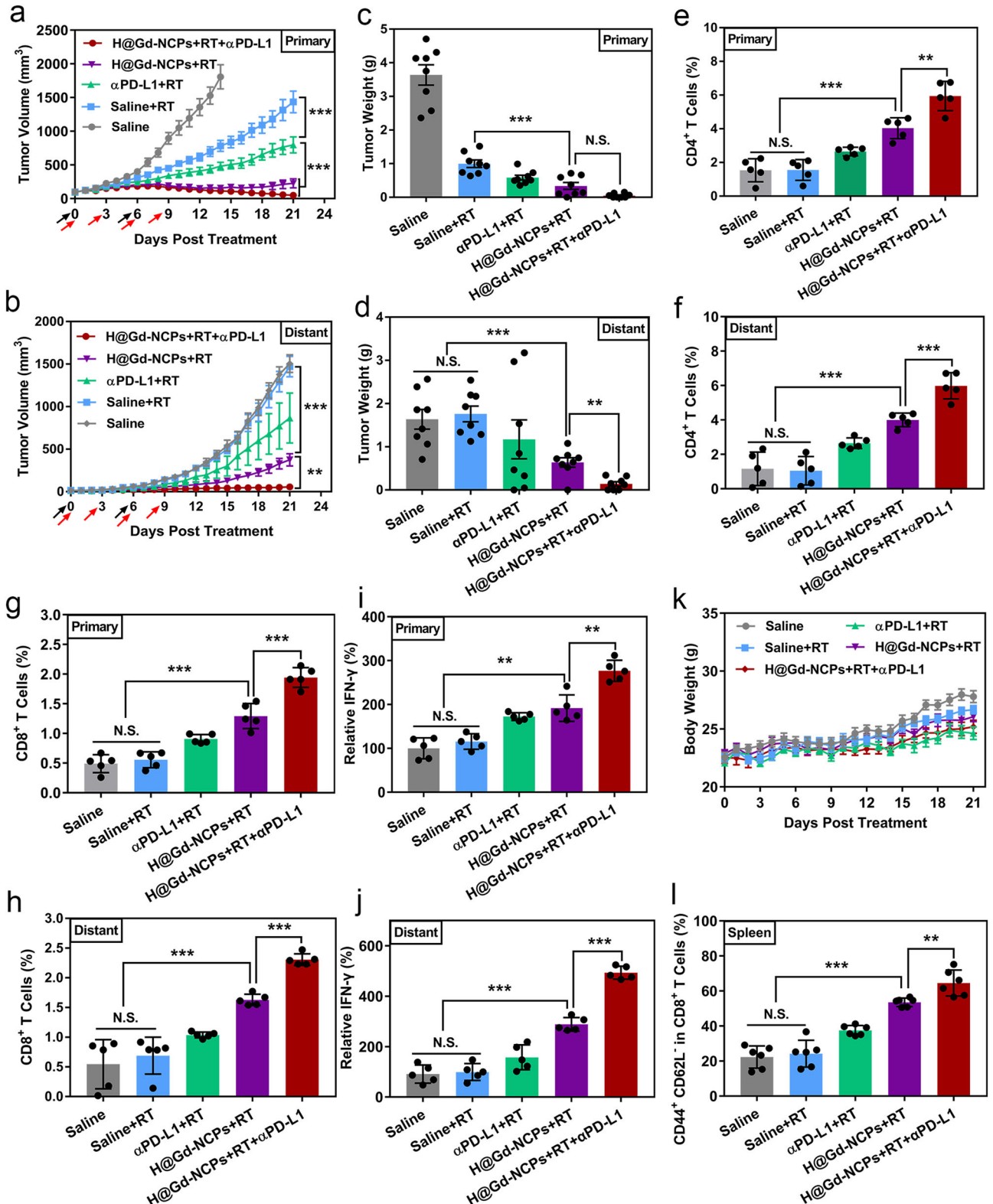

(Supplementary Fig. 25). These results indicated that TAMs might not play a major part in H@Gd-NCPs+RT mediated anti-tumor effects.

**Inhibition of the metastasis of 4T1 breast tumors.** Since H@Gd-NCPs sensitized radiation could effectively induce ICD and activate immune responses against primary and distant CT26 colorectal tumors, we went on to determine the anti-metastatic

effect in a spontaneous 4T1 metastatic breast cancer model (Fig. 9a). Compared with radiation therapy alone, H@Gd-NCPs sensitized radiation enhanced the primary tumor therapeutic efficacy. Similarly, the optimal tumor inhibition was observed after the synergy with CBI by anti-CTLA-4 antibody (αCTLA-4) (Fig. 9b–d and Supplementary Figs. 26, 27). Fourteen days after treatment, primary 4T1 tumors and adjacent skin tissues were carefully dissected. As shown in Fig. 9e, f and Supplementary

**Fig. 7 Abscopal effect and systemic antitumor immunity of H@Gd-NCPs-sensitized RT synergizes with immune checkpoint blockade in vivo. a** Primary (***$p = 0.0001$, ***$p = 0.0001$) and **b** distant (***$p = 0.0001$, **$p = 0.002$) tumor growth curves of CT26 colorectal bilateral tumor-bearing mice treated with Saline, Saline + RT, αPD-L1 + RT, H@Gd-NCPs+RT and H@Gd-NCPs+RT + αPD-L1. [$Gd^{3+}$] = 30 mg $kg^{-1}$, [Hemin] = 12.5 mg $kg^{-1}$ and [αPD-L1] = 10 mg $kg^{-1}$. Treatments were performed on days 0 and 6. X-ray radiation therapy was performed 6 h after nanomedicines intravenous injection (black arrow). RT 6 Gy × 2 with fractions delivered 6 days apart and only primary tumors received radiation therapy. Anti-PD-L1 antibody was treated via intraperitoneal injection 6 h after radiation therapy (red arrow, $n = 8$ biologically independent animals). Data (**a**, **b**) were presented as mean ± SEM. **c** Primary (***$p = 0.0006$) and **d** distant (***$p = 0.0001$, **$p = 0.0012$) CT26 tumors weight ($n = 8$ biologically independent animals). **e**, **f** The percentages of CD4+ T cells in the **e** primary (***$p = 0.0003$, **$p = 0.0039$) and **f** distant (***$p = 0.0003$, ***$p = 0.0009$) tumors analyzed with flow cytometry ($n = 5$ biologically independent animals). **g**, **h** The percentages of CD8+ T cells in the **g** primary (***$p = 0.0001$, ***$p = 0.0006$) and **h** distant (***$p = 0.0001$, ***$p = 0.0001$) tumors analyzed by flow cytometry ($n = 5$ biologically independent animals). **i**, **j** Relative content of IFN-γ in the **i** primary (**$p = 0.0012$, **$p = 0.0011$) and **j** distant (***$p = 0.0001$, ***$p = 0.0001$) tumors detected with ELISA kit ($n = 5$ biologically independent animals), this experiment was repeated twice independently with similar results. Tumor tissues (**e–j**) in different groups were harvested on day 21 for flow cytometry and IFN-γ analysis. **k** Dynamic body weight of bilateral CT26-bearing mice in different groups during treatments ($n = 8$ biologically independent animals). (**l**) Percentages of effector memory T cells ($T_{EM}$) in the spleen analyzed by flow cytometry, the spleen in different groups were collected on day 21 ($n = 6$ biologically independent animals, ***$p = 0.0001$, **$p = 0.0062$). Data (**c–l**) were presented as mean ± SD. Two-sided Student's $t$-test was used to calculate the statistical difference between two groups. N.S. represented non-significance, and **$p < 0.01$, ***$p < 0.001$. Source data are provided as a Source data file.

Fig. 28, the body weight of treated mice significantly decreased before death, and metastatic burdens in the normal tissues were recorded. Notably, the long-term survival rate of H@Gd-NCPs + RT treatment was 30%, significantly higher than that of Saline, RT, and αCTLA-4 + RT groups (0%, 0%, and 10%, respectively). In addition, the synergy with αCTLA-4 further increased the long-term survival rate of H@Gd-NCPs + RT to 50% in 180 days.

The sporadic metastatic lesions could be found within lung tissues in H@Gd-NCPs + RT and H@Gd-NCPs + RT + αCTLA-4 groups, whereas plentiful tumor burdens were observed in other groups (Fig. 9g, h). Similarly, there were fewer metastatic lesions within the livers in H@Gd-NCPs+RT and H@Gd-NCPs + RT + αCTLA-4 groups than in other treated groups (Fig. 9g, i). Furthermore, H&E staining showed that extensive areas of metastatic lesions with agminated nucleus were distributed in the lungs and livers of all groups except H@Gd-NCPs + RT + αCTLA-4 group (Fig. 9j). Collectively, these results demonstrated that H@Gd-NCPs sensitized irradiation could activate immune responses and synergize with CBI to suppress primary and metastatic tumors.

## Discussion

Inducing tumor cells to undergo immunogenic death could effectively elicit systemic anti-tumor immune responses, potentiate CBI, and subsequently improve the survival of tumor patients[1–8]. Although RT can induce ICD to some extent[21,22], it is unsatisfactory due to endogenous resistance mechanisms[29] and poor X-ray absorption by tumor tissues[30]. We thus constructed the NCPs based on gadolinium and 5′-GMP via supramolecular self-assembly, and integrated Hemin with peroxidase-mimic catalytic activity into Gd-NCPs through the coordination interaction to form novel radiosensitizer H@Gd-NCPs. Prof. Qu and co-workers constructed coordination polymer nanoparticles through the self-assembly of GMP and lanthanide ions, such as $Eu^{3+}$. N-methylmesoporphyrin IX (NMM) was confined by π-π stacking in the nanoscale adaptive supramolecular networks (NMM@$Eu^{3+}$/5′-GMP)[52]. Hemin (Iron protoporphyrin IX) and NMM (N-methylmesoporphyrin IX) exhibited very similar structures and properties. Then we speculated that our established H@Gd-NCPs would exhibit a similar structure with NMM@$Eu^{3+}$/5′-GMP, and Hemin was probably encapsulated in the large ring formed by $Gd^{3+}$ and 5′-GMP via π-π stacking. Furthermore, we demonstrated that H@Gd-NCPs could not only effectively synergize radiation and amplify cellular oxidative stress to overcome the resistance to RT, but also induce potent ICD and potentiate CBI.

Recently, nanomaterials based on High-Z elements such as $HfO_2$ nanoparticles (NBTXR3)[53], Gd-based nanoparticles (AGuIX)[54], and nanoscale metal-organic frameworks (RiMO-301)[55,56] have been introduced to the clinic as radiosensitizers. However, the sensibilization effects of these nanomedicines to RT are only unilaterally achieved by depositing X-rays and their clinical benefits are restricted to a certain extent. Besides, intra-tumoral administration of some nanomedicines also severely limited their applications in different types of tumors[53–56]. The established Gd-NCPs not just have similar properties to NBTXR3, AGuIX and RiMO-301, and could effectively deposit X-rays to release a variety of electronics (such as Photoelectron, Putton electronics, Auger electronics)[57], which react with organic molecules or water within tumor cells to generate considerable free radicals and sensitize radiation therapy (Fig. 2h). Notably, Gd-based coordination molecules such as Gd-DTPA (Magnevist, 1988), Gd-DOTA (1989), and Gd-DTPA-BMA (1993) had been approved as MRI contrast agents by FDA[58], indicating bio-compatibility and biosafety of Gd-based coordination polymers. Replacing the exogenous ligands with 5′-GMP to form NCPs could also present MRI capability that outperforms Magnevist in vitro and in vivo (Fig. 4). In addition, body weight recording, serum biochemical analysis, and histological analysis confirmed that Gd-NCPs have biological safety (Figs. 5c, 7k, 9e and Supplementary Figs. 14, 15).

Then, we further investigated the stability of Gd-NCPs and H@Gd-NCPs at an acidic pH. We adjusted the pH of $GdCl_3$, Gd-NCPs, and H@Gd-NCPs solutions to 7.4, 6.5, 5.0, 4.0, 3.0, 2.0, respectively, incubated for 7 days, and then added with TC to detect free $Gd^{3+}$. As shown in Supplementary Fig. 29, Gd-NCPs and H@Gd-NCPs maintained the coordination state at neutral and weak acidic (pH > 4.0), while $Gd^{3+}$ in $GdCl_3$ solution could be easily detected at pH 2.0~7.4. When the pH value was further adjusted to below 3.0, $Gd^{3+}$ could gradually release from Gd-NCPs and H@Gd-NCPs for TC detection. These results potentially indicated that Gd-NCPs and H@Gd-NCPs could maintain the coordination state in the blood circulation, tumor micro-environment, and cell lysosomes. We speculated that the coordination state of Gd in Gd-NCPs and H@Gd-NCPs was highly related to the pKa of 5′-GMP (pKa1 = 2.4). Theoretically, when pH > 7.0, Gd-NCPs or H@Gd-NCPs maintained a relatively stable particulate state. As the pH value gradually decreased, the phosphate was partially mono-protonated (4.0 < pH < 6.0), these nanoparticles would still maintain their particulate or coordination state. When pH < 3.0, free $Gd^{3+}$ could be gradually released from Gd-NCPs or H@Gd-NCPs because of the further proto-nation of phosphate groups (Supplementary Fig. 30). Through

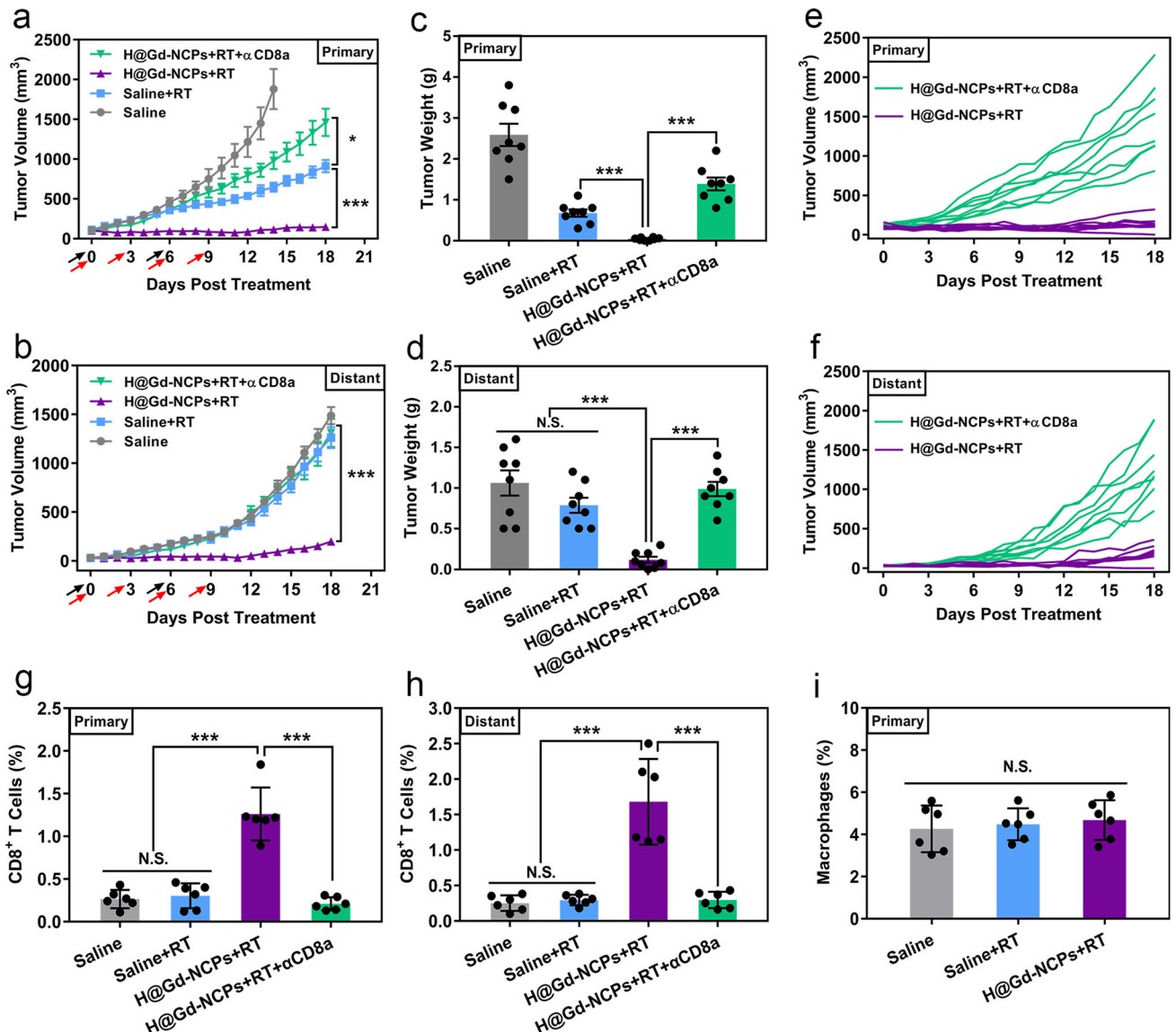

**Fig. 8 CD8$^+$ T cells depletion experiments and ex vivo analysis of immune cells. a** Primary (*$p$ = 0.0345, ***$p$ = 0.0001) and **b** distant (***$p$ = 0.0001) tumor growth curves of CT26 colorectal bilateral tumor-bearing mice treated with Saline, Saline+RT, H@Gd-NCPs+RT and H@Gd-NCPs+RT + αCD8a, ($n$ = 8 biologically independent animals). [Gd$^{3+}$] = 30 mg kg$^{-1}$, [Hemin] =12.5 mg kg$^{-1}$ and [αCD8a] = 10 mg kg$^{-1}$. Treatments were performed on days 0 and 6. X-ray radiation therapy was performed 6 h after nanomedicines intravenous injection (black arrow). RT 6 Gy × 2 with fractions delivered 6 days apart and only primary tumors received radiation therapy. Anti-CD8a antibody was treated via intraperitoneal injection 6 h after radiation therapy (red arrow). Data (**a**, **b**) were presented as mean ± SEM. **c** Primary (***$p$ = 0.0001, ***$p$ = 0.0001) and **d** distant (***$p$ = 0.0001, ***$p$ = 0.0001) CT26 tumor weight ($n$ = 8 biologically independent animals). **e**, **f** Growth curves of **e** primary and **f** distant individual tumors in the H@Gd-NCPs + RT and H@Gd-NCPs + RT + αCD8a groups. **g**, **h** The percentages of CD8$^+$ T cells in the **g** primary (***$p$ = 0.0001, ***$p$ = 0.0001) and **h** distant (***$p$ = 0.0002, ***$p$ = 0.0003) tumors analyzed by flow cytometry ($n$ = 6 biologically independent animals). **i** The percentages of macrophages (F4/80$^+$ and CD11b$^+$) in the primary tumors analyzed by flow cytometry ($n$ = 6 biologically independent animals). Tumor tissues (**g–i**) in different groups were harvested on day 18 for flow cytometry analysis. Data (**c**, **d**, **g–i**) were presented as mean ± SD. Two-sided Student's $t$-test was used to calculate the statistical difference between two groups. N.S. represented non-significance, and *$p$ < 0.05, ***$p$ < 0.001. Source data are provided as a Source data file.

our in vitro simulation studies, we speculated that H@Gd-NCPs would gradually disintegrate into particulate or coordination state, but not free state, after intravenous administration. If a large amount of Gd$^{3+}$ was released, it might cause obvious damages to normal tissues including kidneys. Therefore, we further evaluated the acute toxicity of H@Gd-NCPs and GdCl$_3$ in healthy *Balb/c* mice. The mice were randomly divided into three groups ($n$ = 3), including Saline, GdCl$_3$ ([Gd$^{3+}$] = 3.0 mg kg$^{-1}$ × 6) and H@Gd-NCPs ([Gd$^{3+}$] = 30.0 mg kg$^{-1}$ × 6, 10 times dose of GdCl$_3$). The mice were intravenously injected every day for 6 days and sacrificed on day 7, respectively. As shown in Supplementary

Fig. 31, mice in GdCl$_3$ group exhibited obvious weight loss, while those in Saline and H@Gd-NCPs groups did not. Serum biochemistry analysis indicated that renal function of the mice in GdCl$_3$ group were probably impaired, but no significant difference appeared between Saline and H@Gd-NCPs groups (Supplementary Fig. 32). Histological changes of kidneys merely occurred in GdCl$_3$ group, including multifocal chronic inflammation and interstitial edema (Supplementary Fig. 33), which potentially indicated the relatively bio-safety of H@Gd-NCPs.

Our results and previous study[59] indicated that the therapeutic efficacy of RT sensitization by depositing X-rays alone with High-

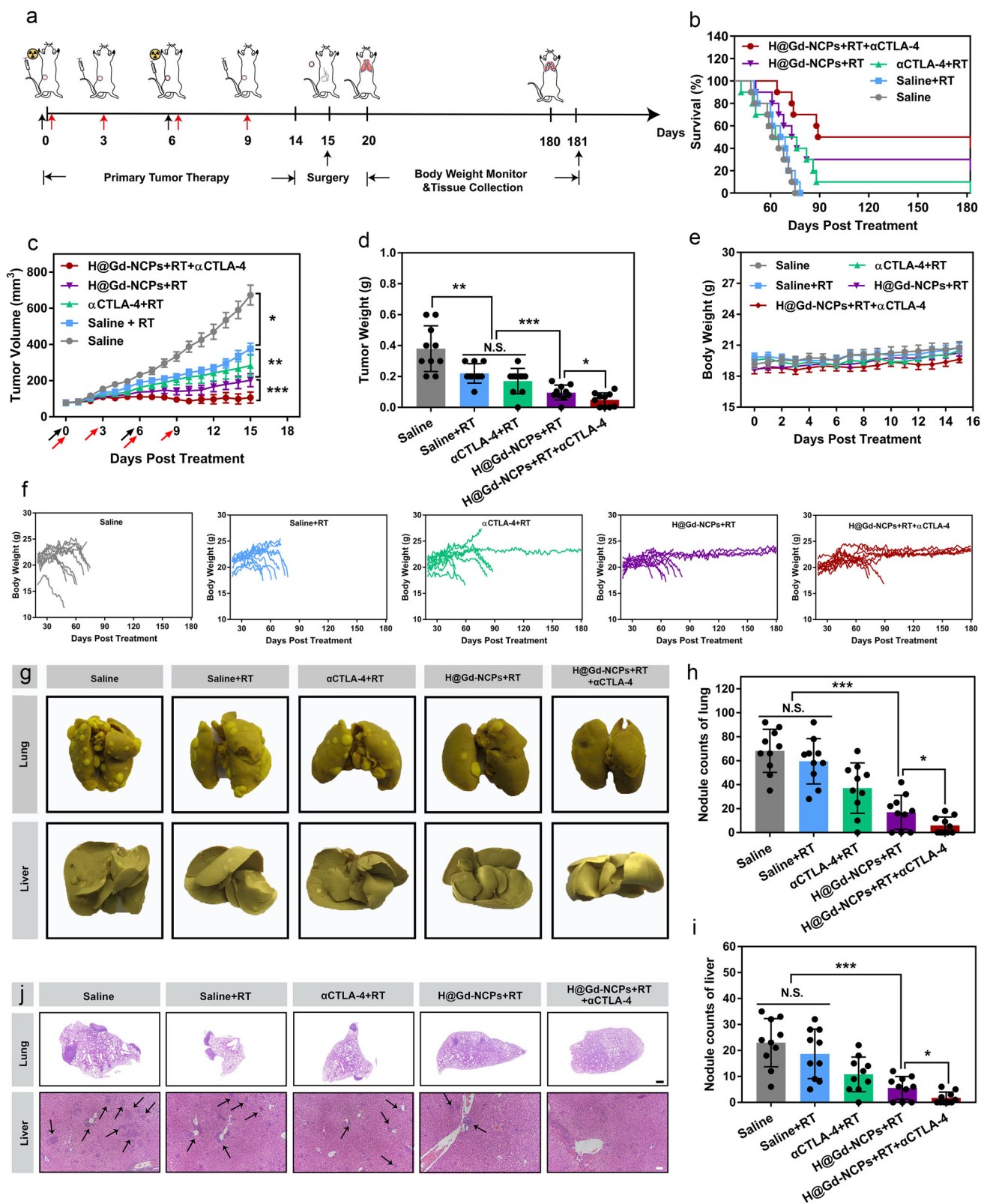

Z element was unsatisfactory (Figs. 2h and 5a). Tumor cells would generate a series of reducing substances to oppose the oxidizing environment, which could greatly weaken the efficacy of radiation by quenching the free radicals. Therefore, we introduced FDA-approved Hemin (PANHEMATIN®) into Gd-NCPs and obtained multi-functional H@Gd-NCPs. During the self-assembly of H@Gd-NCPs, Hemin was uniformly integrated into

Gd-NCPs via its carboxyl groups and conjugated π bonds, effectively avoiding the self-stacking and maintaining catalytic activity. Actually, Hemin could be found in blood and meat products and is widely used as oral iron supplement[60]. With the help of $H_2O_2$, H@Gd-NCPs enhanced the peroxidase-like enzymatic activity of Hemin and exhibited better GSH depletion ability than free Hemin (Fig. 2i), probably due to their uniform

**Fig. 9 Inhibition of primary tumors and metastasis in 4T1 breast tumors. a** Schematic illustration of tumor therapeutic profile. **b** Survival curves of 4T1 breast tumor-bearing mice. **c** Primary tumor growth curves of 4T1 breast tumor-bearing mice treated with Saline, Saline + RT, αCTLA-4 + RT, H@Gd-NCPs + RT and H@Gd-NCPs + RT + αCTLA-4, ($n = 10$ biologically independent animals, $*p = 0.0465$, $**p = 0.0070$, $***p = 0.0003$). [Gd$^{3+}$] = 30 mg kg$^{-1}$, [Hemin] = 12.5 mg kg$^{-1}$ and [αCTLA-4] = 25 μg mouse$^{-1}$. Treatments were performed on days 0 and 6. X-ray radiation therapy was performed 6 h after nanomedicines intravenous injection. RT 6 Gy × 2 with fractions delivered 6 days apart (black arrow). Anti-CTLA-4 antibody was treated via intravenous injection 6 h after radiation therapy (red arrow). Data were presented as mean ± SEM. **d** Primary excised 4T1 breast tumors weight ($n = 10$ biologically independent animals, $**p = 0.0055$, $***p = 0.0007$, $*p = 0.0366$). **e** Dynamic body weights of 4T1 breast tumor-bearing mice ($n = 10$ biologically independent animals). **f** Body weight changes of individual 4T1 breast tumor-bearing mice. **g** Images of lung and liver fixed by Bouin's solution. **h**, **i** Quantification of metastatic lesions of the **h** lungs ($***p = 0.0001$, $*p = 0.04$) and **i** livers ($***p = 0.0009$, $*p = 0.0222$) ($n = 10$ biologically independent animals). **j** H&E sections of lungs (scale bar = 1 mm) and livers (scale bar = 200 μm). Data (**d**, **e**, **h**, **i**) were presented as mean ± SD. Two-sided Student's $t$-test was used to calculate the statistical difference between two groups. N.S. represented non-significance, and $*p < 0.05$, $**p < 0.01$, $***p < 0.001$. Source data are provided as a Source data file.

dispersion. Furthermore, H@Gd-NCPs significantly sensitized radiation therapy by amplifying cellular oxidative stress in vitro and in vivo (Figs. 3 and 5). Enhanced oxidative stress then caused CRT exposure and the release of DAMPs including HMGB1 (Fig. 6), which would then activate DCs for systemic anti-tumor immune responses. CD8$^+$ T cell infiltration in tumor tissues is a vital marker to characterize whether immune checkpoint inhibitors (αPD-1 or αPD-L1) could respond[12]. After H@Gd-NCPs sensitized radiation, CD4$^+$, and CD8$^+$ T cell infiltration significantly increased in both irradiated and non-irradiated tumors, potentiated immune checkpoint inhibitors, and extended radiation therapeutic effects to distant and metastatic tumors for long-term survival (Figs. 7 and 8). When the enhanced infiltration of CD4$^+$ T cells in the bilateral tumor model was observed, we realized that Tregs (regulatory T cells, occupied 20~30% CD4$^+$ T cells in tumor tissues) might play a role in hindering the immune response in the tumor microenvironment[61–65]. Subsequently, in the 4T1 metastatic breast cancer model, we then synergized with the Treg-cell targeting antibody αCTLA-4, which could also obviously extend the survival of mice treated by H@Gd-NCPs + RT (Fig. 9). Therefore, we cautiously speculated that oxidative stress amplification mediated immune stimulation might be impeded by the infiltrated regulatory T cells.

Our established H@Gd-NCPs not only took into account of the X-ray deposition, but also had the function of GSH depletion and Magnetic Resonance Imaging. Moreover, the synergetic therapeutic effects of H@Gd-NCPs could induce powerful ICD and potentiate CBI for systemic anti-tumor immunity. In addition, Gd-NCPs and H@Gd-NCPs could be freeze-dried and the lyophilized powders could be stored for later use (Supplementary Fig. 34). Furthermore, other therapeutical molecules, including chemotherapeutic drugs (doxorubicin, mitoxantrone), photosensitizer (Alcian Blue 8GX) and photothermal agent (ICG) could be encapsulated within Gd-NCPs, which could be a promising in vivo delivery platform (Supplementary Figs. 35 and 36).

In summary, we developed H@Gd-NCPs based on peroxidase mimic Hemin, High-Z gadolinium, and endogenous 5′-GMP as a radiosensitizer, which could not only be used for MRI, but also induce potent ICD and potentiate CBI for systemic anti-tumor immunity. The established H@Gd-NCPs with biocompatibility and therapeutic outcomes potentially provide a novel approach for radiosensitization and anti-tumor immunotherapy.

## Methods

**Materials**. Gadolinium trichloride hexahydrate and Hemin were purchased from Energy Chemical (China). Guanosine-5′-monophosphate disodium salt was supplied by Yuanye (China). 4′,6-diamidino-2-phenylindole (DAPI) and Lyso-tracker Green were obtained from Sigma-Aldrich (USA). GSH and GSSG Assay Kit and ATP Assay Kit were purchased from Beyotime Biotech Inc (China). 2′,7′-Dichlorodihydrofluorescein diacetate (H$_2$DCFDA) was supplied by KeyGEN Biotech (China). Cell counting kit-8 (CCK-8) was obtained from Dojindo Laboratories (Japan). Gadopentetate dimeglumine (Magnevist) was purchased from HEOWNS

Biochem LLC (China). Mouse IFN-γ precoated ELISA Kit was obtained from DAKEWEI (China). Mouse HMGB1 ELISA Kit was obtained from Yifeixue BIO THCHNOLOGY (China). Anti-HMGB1 antibody (Cat# ab18256, diluted 1:300 with 3% BSA), Anti-Calreticulin antibody [EPR3924]—ER Marker (Alexa Fluor® 488) (Cat# ab196158, diluted 1:500 with 3% BSA), Anti-gamma H2A.χ (phospho S139) antibody [9F3] (Cat# ab26350, diluted 1:200 with 3% BSA), Goat Anti-Mouse IgG H&L (Alexa Fluor® 488) (Cat# ab150113, diluted 1:200 with 3% BSA), Anti-Ki67 antibody (Cat# ab15580, diluted 1:500 with 3% BSA), Goat Anti-Mouse IgG H&L (HRP) (Cat# ab6789, diluted 1:300 with 3% BSA), TUNEL Assay Kit-BrdU-Red (Cat# ab66110, diluted 1:200 with 3% BSA), Anti-CD4 antibody [EPR19514] (Cat# ab183685, diluted 1:400 with 3% BSA), Anti-CD8 alpha antibody [EPR21769] (Cat# ab217344, diluted 1:400 with 3% BSA) and Goat Anti-Mouse IgG H&L (Cy3®) preadsorbed (Cat# ab97035, diluted 1:300 with 3% BSA) were purchased from Abcam (USA). FITC anti-mouse CD11c Antibody [N418] (Cat# 117306, 0.25 μg per million cells in 100 μL volume), APC anti-mouse CD80 Antibody [16-10A1] (Cat# 104713, 1.0 μg per million cells in 100 μL volume), PE anti-mouse CD86 Antibody [GL-1] (Cat# 105007, 0.25 μg per million cells in 100 μL volume), APC anti-mouse CD3 Antibody [17A2] (Cat# 100236, 0.5 μg per million cells in 100 μL volume), PE anti-mouse CD4 Antibody [GK1.5] (Cat# 100408, 0.25 μg per million cells in 100 μL volume), FITC anti-mouse CD8a Antibody [53-6.7] (Cat# 100706, 1.0 μg per million cells in 100 μL volume), PE anti-mouse CD3 Antibody [17A2] (Cat# 100205, 0.5 μg per million cells in 100 μL volume), APC anti-mouse/human CD44 Antibody [IM7] (Cat# 103011, 0.25 μg per million cells in 100 μL volume), PerCP/Cyanine5.5 anti-mouse CD62L Antibody [MEL-14] (Cat# 104432, 0.25 μg per million cells in 100 μL volume), PE anti-mouse F4/80 Antibody [BM8] (Cat# 123110, 1.0 μg per million cells in 100 μL volume), PerCP/Cyanine5.5 anti-mouse/human CD11b Antibody [M1-70] (Cat# 101227, 0.25 μg per million cells in 100 μL volume) and Ultra-LEAF™ Purified anti-mouse CD8a Antibody [53-6.7] (Cat# 100764) were purchased from BioLegend (USA). In vivo MAb anti-mouse PD-L1(B7-H1) (Clone: 10 F.9G2, Cat# BE0101) and In vivo MAb anti-mouse CTLA-4 (CD152) (Clone: UC10-4F10-11, Cat# BE0032) were purchased from BioXcell (USA).

**Cell lines**. The mouse CT26 colorectal cancer cells and 4T1 breast cancer cells were purchased from China Type Culture Collection, supplied by the American Type Culture Collection.

**Animals**. *Balb/c* mice were obtained from the Yangzhou University Medical Centre (Yangzhou, China). All animal work was approved by the Institution Animal Care and Use Committee (IACUC) of Nanjing University and was conducted in accordance with the principles of the Association for Assessment and Accreditation of Laboratory Animal Care International (AAALAC). *Balb/c* mice (Male, 5 weeks) for the construction of CT26-bearing mice and *Balb/c* mice (Female, 5 weeks) for the construction of 4T1-bearing mice. The animals were hosted in an equipped animal facility with temperature at 20–25 °C and humidity at 30%–70%, under the same dark/light cycle (12:12).

**Software**. All tumor size and mice body weight were recorded by Microsoft Office 2019. Sante MRI Viewer 3.0 was used to analyze MRI data. GraphPad Prism Version 7.0 was used to analyze statistical data. FlowJo Version 7.6.1 was used to analyze flow cytometry data. NIS-Elements Viewer 5.21.00 and ImageJ Version 1.52v were used to analyze immunofluorescent and immunochemical data.

**Preparation and characterization of Gd-NCPs and H@Gd-NCPs**. Preparation of H@Gd-NCPs was typically conducted as follows[43]. First, the aqueous solution of 10 mM 5′-GMP disodium salt (30 mL) was mixed with 1 mM Hemin (20 mL), and stirred for 30 min. Then, the aqueous 10 mM solution of GdCl$_3$ (20 mL) was added to the mixed solution slowly at room temperature. After 1 h of coordination, the precipitates were collected by centrifugation (4000 × $g$, 20 min). The obtained precipitate was washed with pure water three times and collected by centrifugation (4000 × $g$, 20 min). Then, ultrasonication was performed to obtain NCPs

H@Gd-NCPs. The Gd-NCPs were prepared with a similar method, without the integration of Hemin. Particle sizes of Gd-NCPs and H@Gd-NCPs were determined by Dynamic Light Scattering (DLS, Brookhaven 90 plus Zeta). The stability of Gd-NCPs and H@Gd-NCPs were also investigated by DLS under different conditions for 48 h including Saline and 50% serum at 25 °C and 37 °C, respectively.

**Quantitative analysis of [$Gd^{3+}$] and Hemin in H@Gd-NCPs.** To determine the quantity of Hemin and [$Gd^{3+}$] in H@Gd-NCPs, the concentration of [$Gd^{3+}$] in the supernatant was determined by colorimetry[43]. TC was used as a colorimetric reagent for free [$Gd^{3+}$]. A calibration curve was obtained by plotting absorbance of TC solution at 605 nm in the presence of various concentration of [$Gd^{3+}$] ([TC] = 250 μM, [$Gd^{3+}$] = 10, 20, 40, 50, 60, 80, 100, 120, 140, 160 μM). In addition, the calibration curve was obtained by plotting absorbance of Hemin aqueous solution ([Hemin] = 10, 20, 30, 40, 50, 60 μM) at 392 nm by ultraviolet spectrophotometer (UV) and Hemin concentration was determined.

**Study on the stability of Gd-NCPs and H@Gd-NCPs at different pH.** Gd-NCPs ([$Gd^{3+}$] = 20 mM, 500 μL) and H@Gd-NCPs ([$Gd^{3+}$] = 20 mM and [Hemin] = 2 mM, 500 μL) were respectively packed into dialysis bags (Solarbio, 10 kD), followed by dialysis in 50% bovine serum (5.0 mL, pH = 7.4) or deionized water (5.0 mL, pH = 7.4, 6.5, 5.0, 4.0, 3.0 or 2.0) for 7 days, respectively. The dialysates were concentrated to 1.0 mL to detect free $Gd^{3+}$ by colorimetry[43].

**Trans-metallation experiments of H@Gd-NCPs in vitro.** In order to evaluate whether the H@Gd-NCPs would undergo trans-metallation after injection, H@Gd-NCPs ([$Gd^{3+}$] = 20 mM and [Hemin] = 2 mM, 1.0 mL) were respectively packed into dialysis bags (Solarbio, 10 kD), stirred in the 100.0 mL dialysate (50% bovine serum, adding extra [$Na^+$] = 150 mM, [$K^+$] = 5.0 mM, [$Ca^{2+}$] = 2.5 mM, [$Mg^{2+}$] = 1.25 mM, [$Zn^{2+}$] = 30 μM, [$Fe^{3+}$] = 30 μM, [$Cu^{2+}$] = 30 μM to mimic physiological environment, pH = 7.4, 6.5 or 5.0) for 7 days, respectively. The dialysates were concentrated by vacuum distillation, and the concentrated residues were analyzed by ICP-OES (Avio 500, USA).

**Evaluation of hydroxyl radical (•OH) generation in vitro.** To detect the generation of hydroxyl radical (•OH) under irradiation, a classical colorimetric method was used based on the decay of MB under the physiological environment. In brief, $H_2O$, $GdCl_3$, Gd-NCPs or H@Gd-NCPs were added into MB solutions ([$Gd^{3+}$] = 20 μM and [MB] = 15 μg mL$^{-1}$). After irradiating for various doses (0, 5 Gy × 1, 10 Gy × 1, 20 Gy × 1), the absorption of MB at 664 nm was measured to detect the degradation of MB.

**Evaluation of GSH depletion in vitro.** Mixed the aqueous solutions of 0.5, 1.0 or 2.0 μM Hemin (400 μL) or H@Gd-NCPs (400 μL), 4 mM GSH (400 μL) and 400 mM $H_2O_2$ (20 μL) and stirred for 1 h and 24 h, respectively. Then the concentrations of GSH were determined by GSH and GSSG Assay Kit (S0053, Beyotime, China).

**Subcellular localization.** CT26 cells were cultured in 1640 medium supplemented with 10% FBS and 1% penicillin/streptomycin under 5% $CO_2$ at 37 °C and they were seeded with a density of 3 × 10$^4$ well$^{-1}$ in 24-well plates covered by glass disk. After attachment, the cells were incubated with H@Gd-NCPs for 4 h. Lysotracker and DAPI were used to label lysosome and nucleus, respectively. The tumor cells were washed three times with PBS, then images were obtained and analyzed by Olympus FV3000 LSCM.

**Intracellular ROS generation.** The ROS generation was detected by $H_2$DCFDA. CT26 cells were seeded in 96-well plates at 8 × 10$^3$ well$^{-1}$ and further cultured for 12 h. Gd-NCPs ([$Gd^{3+}$] = 100 μM) and H@Gd-NCPs were added to the cells ([$Gd^{3+}$] = 100 μM, [Hemin] = 10 μM) and then transferred into the anaerobic chamber for another 12 h. Radiation was performed with the dose of 8 Gy × 1. According to the protocol, 10 μM $H_2$DCFDA (KeyGen Biotech, China) was added and incubated for 1 h. Then the $H_2$DCFDA-loading medium was replaced with fresh medium and incubated for another 30 min. Then, the cells were washed with fresh medium twice. Tumor cells without radiation were used as control. Immunofluorescence images were obtained from Nikon Eclipse Ti (Japan) and analyzed with ImageJ Software.

**Intracellular GSH/GSSG ratio detection.** The intracellular GSH was detected using GSH and GSSG Assay Kit (Beyotime, China). CT26 cells were seeded in 6-well plates at 2 × 10$^5$ well$^{-1}$ and further cultured for 12 h. Gd-NCPs ([$Gd^{3+}$] = 100 μM) or H@Gd-NCPs ([$Gd^{3+}$] = 100 μM, [Hemin] = 10 μM) and $H_2O_2$ (100 μM) were respectively added to the cells and then transferred into the anaerobic chamber (5% $CO_2$, 93% $N_2$ and 2% $O_2$) for another 12 h. The treated tumor cells were lysed by repeated cycle of freezing and thawing, then centrifuged to collect the supernatant for the measurement of GSH and GSSG according to the manufacturer's protocol.

**Cytotoxicity.** The cytotoxicity was evaluated with CCK-8 assay (Dojindo Laboratories, Japan). To investigate the cytotoxicity of Gd-NCPs and H@Gd-NCPs with or without radiation, CT26 cells were seeded on 96-well plates at 5 × 10$^3$ well$^{-1}$ and further cultured for 12 h. Then, Gd-NCPs and H@Gd-NCPs ([$Gd^{3+}$] = 0, 12.5, 25, 50, and 100 μM) and $H_2O_2$ (100 μM) were added and incubated in an anaerobic chamber (5% $CO_2$, 93% $N_2$ and 2% $O_2$) for 12 h. After radiation (8 Gy × 1), CT26 cells were cultured continuously for another 72 h (culture medium was refreshed every 24 h) before determining the cell viability by CCK-8 assay. The cytotoxicity of Gd-NCPs and H@Gd-NCPs ([$Gd^{3+}$] = 0, 12.5, 25, 50, 100, and 200 μM) without radiation was performed with similar methods as negative control.

**Cell clonogenic assay.** CT26 cells were treated as cell cytotoxicity experiments mentioned. Two hours after the radiation (8 Gy × 1), CT26 cells of all groups (Saline, Saline + RT, Gd-NCPs + RT, H@Gd-NCPs + RT, n = 3) were transferred into six-well plates at densities of 2000 cells per well and cultivated for another 7 days. Once the microscopic cell colonies were formed, cells of different groups were fixed with anhydrous ethanol and stained with crystal violet. The number of surviving colonies were counted and this experiment was repeated twice independently with similar results.

**CRT exposure analysis.** For detection of CRT exposure, CT26 cells were seeded on glass coverslip placed in 24-well plates at a density of 10$^5$ cells well$^{-1}$ and cultured for 12 h. Then, Gd-NCPs ([$Gd^{3+}$] = 100 μM) and H@Gd-NCPs ([$Gd^{3+}$] = 100 μM, [Hemin] = 10 μM) were added to the cells and then transferred into the anaerobic chamber for another 12 h. After radiation (8 Gy × 1), CT26 cells were cultured continuously for another 4 h and washed with PBS for three times. Then, tumor cells were incubated with Alexa Fluor® 488-CRT antibody (diluted 1:500 with 3% BSA, Abcam, USA) for 1 h, stained with DAPI. Immunofluorescence images were obtained from Nikon Eclipse Ti (Japan) and analyzed with ImageJ Software.

**Detection of HMGB1 and ATP release.** HMGB1 concentrations in the cytoplasm of cells following the indicated treatments were measured by ELISA kit (Yifeixue Bio, China), according to the manufacturer's protocol. ATP concentrations in the supernatant of cells upon the indicated treatments were measured by ATP Assay Kit (Beyotime, China), according to the manufacturer's protocol. Luminescence and absorbance were measured by using microplate reader (VICTOR® Nivo).

**Accumulation and MR imaging in vivo.** All the animals were obtained from Yangzhou university medical center (Yangzhou, China) and received care in accordance with the Institution Animal Care and Use Committee (IACUC) of Nanjing University. To evaluate the accumulation of H@Gd-NCPs in tumor tissues and imaging ability of [$Gd^{3+}$], CT26 tumor-bearing mice (150–200 mm$^3$) were treated with H@Gd-NCPs with intravenous injection. Mice were anesthetized with isoflurane and fixed in the animal groove. $T_1$-weighted images were carried out on MR scanner (Biospec 7T/20 USR, Germany). Parameters used for T1-weighted imaging were as follows: flip angle = 180, TR = 500 ms, TE = 15.0 ms, FOV = 3 × 3, matrix = 256 × 256, SI = 1.0 mm 1.0 mm$^{-1}$, averages = 3, slices = 12, NEX = 1. Multiple locations of phantom images were observed at 0, 2, 6, 12, 24, 48, and 60 h and analyzed with ImageJ Software and Nikon NIS-Elements.

**Pharmacokinetic study of H@Gd-NCPs and Magnevist.** All the animals were obtained from Yangzhou university medical center (Yangzhou, China) and received care in accordance with the Institution Animal Care and Use Committee (IACUC) of Nanjing University. To evaluate the absolute quantification of H@Gd-NCPs and Magnevist in tumor tissues, CT26 tumor-bearing mice (150–200 mm$^3$) were treated with H@Gd-NCPs ([$Gd^{3+}$] = 30 mg kg$^{-1}$ and [Hemin] = 12.5 mg kg$^{-1}$) or Magnevist ([$Gd^{3+}$] = 30 mg kg$^{-1}$) with intravenous injection. Tumor tissues (0.2 g) were respectively collected at 0, 2, 6, 12, 24, 48, and 60 h after mice sacrificed (n = 3). Then, the tumor tissues were crushed and homogenized to 2.0 mL, followed by centrifuged (4000 × g, 10 min) and the supernatant was carefully collected and divided into two equal parts, one part was directly (1.0 mL) to quantify [$Gd^{3+}$], the other part was burned, nitrified and then re-diluted to 1.0 mL to quantify [$Gd^{3+}$] via colorimetry[43].

**Serum dilution experiments of H@Gd-NCPs in vitro.** To testify that the size of H@Gd-NCPs could be degraded in the process of blood circulation, H@Gd-NCPs ([$Gd^{3+}$] = 20 mM and [Hemin] = 2 mM) were serial diluted by 50% bovine serum at 37 °C, and the dilution ratios were ×16, ×32, ×64, ×128, ×256, and ×512 respectively. Then, the size of H@Gd-NCPs in different concentrations were respectively determined by Dynamic Light Scattering (DLS, Brookhaven 90 plus Zeta).

**Mice urine collection and analysis.** All the animals were obtained from Yangzhou university medical center (Yangzhou, China) and received care in accordance with the Institution Animal Care and Use Committee (IACUC) of Nanjing University. To evaluate the final distribution of the gadolinium, CT26 tumor-bearing mice (100–120 mm$^3$) were treated with H@Gd-NCPs ([$Gd^{3+}$] = 30 mg kg$^{-1}$ and

[Hemin] = 12.5 mg kg$^{-1}$) with intravenous injection. Mice urine from the bladder was collected at 24~48 h after mice sacrificed ($n$ = 3 biologically independent animals). One part of the urine was directly used for free Gd$^{3+}$ detection, another part of the urine was burned and nitrified for Gd$^{3+}$ detection.

**Enhanced radiation therapy in CT26-bearing mice.** CT26 tumor-bearing mice (110–130 mm$^3$) were divided into six groups, which were treated with Saline, Gd-NCPs ([Gd$^{3+}$] = 30 mg kg$^{-1}$) and H@Gd-NCPs ([Gd$^{3+}$] = 30 mg kg$^{-1}$ and [Hemin] = 12.5 mg kg$^{-1}$) with or without radiation, respectively. Treatments were performed on days 0 and 6. X-ray radiation therapy was performed 6 h after nanomedicine intravenous injection. RT 6 Gy × 2 with fractions delivered 6 days apart. The tumor size and body weight were recorded every day, and tumor volume was calculated according to the formula: width$^2$ × length/2. To inspect the biocompatibility, blood for serum biochemistry was collected before sacrifice. Serum biochemistry data were measured using Chemray 430 (Rayto, China). Hematoxylin and eosin (H&E) staining of major organs and tumors were performed on day 21.

**Immunofluorescence and Immunochemistry.** γ-H2Aχ antibody, TUNEL assay kit, and Ki67 antibody were used for the staining of DNA double-strand breaks, apoptosis and proliferative cells, respectively. Mice bearing CT26 (150~200 mm$^3$) were treated with different samples via intravenous injection 6 h prior to RT (6 Gy × 1). Tumor slices of γ-H2Aχ were obtained from sacrificed mice 24 h after radiation. Then, γ-H2Aχ rabbit monoclonal primary antibody (diluted 1:200 with 3% BSA, Abcam, USA) and a secondary antibody conjugated with Alexa Fluor 488 (diluted 1:200 with 3% BSA, Abcam, USA) were applied to estimate DNA double-strand breaks. Tumor slices of Ki67 and TUNEL were obtained from sacrificed mice 48 h after radiation. For Ki67 immunochemistry, primary rabbit Ki67 antibody (diluted 1:500 with 3% BSA, Abcam, USA) and second goat anti-rabbit antibody conjugated with horseradish peroxidase (HRP) (diluted 1:300 with 3% BSA, Abcam, USA) were used to investigate proliferative cells in the tumor tissues. For TUNEL immunofluorescence, manufacturer's instructions were followed for all the procedures for TUNEL assay (diluted 1:200 with 3% BSA, Abcam, USA). Immunofluorescence and immunochemistry images were obtained using Nikon Eclipse Ti (Japan) and analyzed with ImageJ Software.

**Western blot of HMGB1.** The total protein in cytoplasm and cell matrix were extracted by Cytoplasmic Protein Extraction Kit (Beyotime Biotech, China). The tumor tissues collected from mice in different groups were crushed via a mechanical tissue homogenizer, and then mixed with cytoplasmic protein extraction reagents with slowly stirring at 4 °C. Cytoplasmic protein extraction reagents could fully swell the cells via low osmotic pressure, destroy the cell membrane, release cytoplasmic proteins, and the nuclear precipitate could be removed by centrifugation (12,000 × g × 10 min, 4 °C). Then, the tumor tissue supernatants were collected for HMGB1 (diluted 1:300 with 3% BSA, Abcam, USA) detection by Western blot.

**Gating strategy and methodology of flow cytometry.** All of the flow cytometry experiments were adopted with this sample treatment method and gating strategy. After incubated with various antibodies, cells were fixed by 4% paraformaldehyde and then analyzed via flow cytometry. During the running process, Forward Scatter (FSC) and Side Scatter (SSC) dot maps were established, the voltage was adjusted to ensure that all the events were within the visible range of the dot maps. Then, the events with appropriate FSC (200–600) and SSC (200–600) were gated and collected. Those events with low FSC/low SSC and low FSC/high SSC were abandoned, which mainly represented cell debris and air bubbles. Followed by cell type-specific gating using fluorescently labeled antibodies. DCs (CD80$^+$ and CD86$^+$ gated on CD11c$^+$); CD4$^+$ T cells (CD3$^+$ and CD4$^+$); CD8$^+$ T cells (CD3$^+$ and CD8$^+$); Effector memory T cells (T$_{EM}$, CD62L$^-$ CD44$^+$ gated on CD3$^+$ CD8$^+$); Macrophages (F4/80$^+$ and CD11b$^+$), respectively.

**In vivo DC cells maturation.** To evaluate DCs maturation, CT26 tumor-bearing mice (110–130 mm$^3$) were divided into six groups, which were treated with Saline, Gd-NCPs ([Gd$^{3+}$] = 30 mg kg$^{-1}$), H@Gd-NCPs ([Gd$^{3+}$] = 30 mg kg$^{-1}$ and [Hemin] = 12.5 mg kg$^{-1}$) with radiation (6 Gy × 1) and Saline, Gd-NCPs, H@Gd-NCPs without radiation, respectively. Five days post radiation, mice were sacrificed and tumor-draining lymph nodes (TDLNs) were harvested for flow cytometry analysis. The TDLNs were ground into single-cell suspension, stained with FITC anti-mouse CD11c (0.25 µg per million cells in 100 µL volume), PE anti-mouse CD86 (0.25 µg per million cells in 100 µL volume), APC anti-mouse CD80 (1.0 µg per million cells in 100 µL volume) antibodies (BioLegend, America) and then detected by flow cytometry (BD Calibur).

**Abscopal effect of H@Gd-NCPs sensitized radiation therapy.** To inspect the function of H@Gd-NCPs in eliciting the abscopal effect of radiation therapy, mice bearing both primary and distant tumors were established. CT26 cells (right: 5 × 10$^5$ cells each mouse, left: 5 × 10$^4$ cells each mouse) were injected subcutaneously in the right and left lower flank of mice to form primary and distant tumors. Ten days later, CT26 tumor-bearing mice (~120 mm$^3$ for primary tumor, ~20 mm$^3$ for

distant tumor) were randomly divided into five groups, including Saline, αPD-L1, H@Gd-NCPs, H@Gd-NCPs+αPD-L1 ([Gd$^{3+}$] = 30 mg kg$^{-1}$ and [Hemin] = 12.5 mg kg$^{-1}$) with radiation and Saline without radiation, respectively. Treatments were performed on days 0 and 6. X-ray radiation therapy was performed 6 h after nanomedicines intravenous injection. RT 6 Gy × 2 with fractions delivered 6 days apart. Only primary tumors received radiation therapy. Anti-PD-L1 antibody (10 mg kg$^{-1}$ × 4 with fractions delivered 3 days apart, Clone: 10 F.9G2, BioXcell, America) was treated via intraperitoneal injection 6 h after radiation therapy. Treatments were performed according to schematic illustration of tumor therapeutic treatments. Both primary and distant tumors were collected on day 21. To inspect the infiltration of immune cells precisely, fresh tumor tissues were detached into single cells with enzyme mixture including neutral protease (3 mg mL$^{-1}$), collagenase II (6 mg mL$^{-1}$), hyaluronidase (75 µg mL$^{-1}$) and analyzed with flow cytometry after staining with APC anti-mouse CD3 (0.5 µg per million cells in 100 µL volume), PE anti-mouse CD4 (0.25 µg per million cells in 100 µL volume) and FITC anti-mouse CD8a (1.0 µg per million cells in 100 µL volume) antibodies (BioLegend, USA). Furthermore, the IFN-γ secreted by active CD8$^+$ T cells were detected with ELISA assay kits (Dakewe Biotech Co., Ltd.) according to the manufacturer's instructions.

**Immunofluorescent and immunochemical staining of CD4$^+$ and CD8$^+$ T cells infiltration in tumor.** To further prove the infiltration of CD4$^+$ and CD8$^+$ T cells were enhanced following H@Gd-NCPs+RT treatments, tumor tissues in different groups were harvested on day 21 for immunofluorescent and immunohistochemical staining of CD4$^+$ and CD8$^+$ T cells. For immunofluorescent staining, tumor tissues were cut into small pieces and coated by paraffin firstly, and then paraffin section were incubated with primary anti-CD4 or anti-CD8 antibodies (diluted 1:400 with 3% BSA, Abcam, USA) overnight in a wet box at 4 °C, relevant secondary antibodies conjugated with Cy3® (diluted 1:300 with 3% BSA, Abcam, USA) were added to incubate for 50 min at 37 °C in darkness. Finally, DAPI solutions were added to incubate for 10 min at 37 °C in darkness for nucleus staining. For immunochemical staining, primary anti-CD4 or anti-CD8 antibodies (diluted 1:400 with 3% BSA, Abcam, USA) were incubated with paraffin section overnight in a wet box at 4 °C, and then relevant secondary antibodies conjugated with HRP (diluted 1:300 with 3% BSA, Abcam, USA) were added to incubate for 50 min at 37 °C in darkness. The chromogenic reagents (Diaminobenzidine and Hematoxylin solutions) were used for nucleus staining. Immunofluorescence and immunochemistry images were obtained from Nikon Eclipse Ti (Japan) and analyzed with ImageJ Software.

**Ex vivo analysis of memory T cells.** PE anti-mouse CD3 (0.5 µg per million cells in 100 µL volume), FITC anti-mouse CD8a (1.0 µg per million cells in 100 µL volume), APC anti-mouse CD44 (0.25 µg per million cells in 100 µL volume) and PerCP/Cy5.5 anti-mouse CD62L (0.25 µg per million cells in 100 µL volume) antibodies (BioLegend, USA) were introduced to identify effector memory T cells (T$_{EM}$, CD3$^+$ CD8$^+$ CD62L$^-$ CD44$^+$) and central memory T cells (T$_{CM}$, CD3$^+$ CD8$^+$ CD62L$^+$ CD44$^+$) by flow cytometry (BD Calibur).

**CD8$^+$ T cells depletion and ex vivo analysis of immune cells.** To assess the role of in vivo CD8$^+$ T cell in anti-tumor therapeutic effects, *Balb/c* mice bearing both primary and distant tumors were established. CT26 cells (right: 5 × 10$^5$ cells each mouse, left: 5 × 10$^4$ cells each mouse) were injected subcutaneously in the right and left lower flank of mice to form primary and distant tumors. Ten days later, CT26 tumor-bearing mice (~100 mm$^3$ for primary tumor, ~30 mm$^3$ for distant tumor) were randomly divided into four groups, including Saline, H@Gd-NCPs, H@Gd-NCPs+αCD8a ([Gd$^{3+}$] = 30 mg kg$^{-1}$ and [Hemin] = 12.5 mg kg$^{-1}$) with radiation and Saline without radiation, respectively. Treatments were performed on days 0 and 6. X-ray radiation therapy was performed 6 h after nanomedicine intravenous injection. RT 6 Gy × 2 with fractions delivered 6 days apart. Only primary tumors received radiation therapy. Ultra-LEAF™ Purified anti-mouse CD8a (10 mg kg$^{-1}$ × 4 with fractions delivered 3 days apart, Clone: 53-6.7, BioLegend, USA) was treated via intraperitoneal injection 6 h after radiation therapy. Treatments were performed according to the schematic illustration of tumor therapeutic treatments. Both primary and distant tumors were collected on day 18. To inspect the infiltration of immune cells precisely, fresh tumor tissues were detached into single cells with enzyme mixture including neutral protease (3 mg mL$^{-1}$), collagenase II (6 mg mL$^{-1}$), hyaluronidase (75 µg mL$^{-1}$) and analyzed with flow cytometry after staining with APC anti-mouse CD3 (0.5 µg per million cells in 100 µL volume) and FITC anti-mouse CD8a (1.0 µg per million cells in 100 µL volume) antibodies (CD8$^+$ T cells); PE anti-mouse F4/80 (1.0 µg per million cells in 100 µL volume) and Percp/Cy5.5 anti-mouse CD11b (0.25 µg per million cells in 100 µL volume) antibodies (Macrophages), respectively.

**Inhibition of tumor metastasis in 4T1-bearing mice.** To evaluate the function of H@Gd-NCPs in inhibiting tumor metastasis, mice bearing 4T1 were established. 4T1 cells (5 × 10$^5$ cells each mouse) were injected subcutaneously in the right lower flank of mouse. Ten days later, 4T1 breast tumor-bearing mice (~80 mm$^3$) were randomly divided into five groups, which was respectively treated with Saline, αCTLA-4, H@Gd-NCPs ([Gd$^{3+}$] = 30 mg kg$^{-1}$ and [Hemin] = 12.5 mg kg$^{-1}$),

H@Gd-NCPs + αCTLA-4 with radiation or Saline without radiation. Treatments were performed on days 0 and 6. X-ray radiation therapy was performed after nanomedicines intravenous injection 6 h. RT 6 Gy × 2 with fractions delivered 6 days apart. Anti-CTLA-4 antibody (25 μg mouse$^{-1}$ × 4 with fractions delivered 3 days apart, Clone: UC10-4F10-11, BioXcell, America) was treated via intravenous injection after radiation therapy 6 h. Treatments were performed according to schematic illustration of tumor therapeutic treatments. After continuous monitoring tumor volume and body weight for 14 days, the primary tumor and surrounding skin tissues were carefully removed in order to minimize the risk of primary tumor relapse. Body weight and survival status were monitored after 3 days of surgery and lasted for the following 180 days.

**Acute toxicity of H@Gd-NCPs and GdCl₃ in vivo**. To further evaluated the relatively bio-safety of H@Gd-NCPs. The healthy *Balb/c* mice were randomly divided into three groups ($n = 3$), including Saline, GdCl₃ ([Gd$^{3+}$] = 3.0 mg kg$^{-1}$ × 6) and H@Gd-NCPs ([Gd$^{3+}$] = 30.0 mg kg$^{-1}$ × 6, 10 times dose of GdCl₃). The mice were intravenously injected every day for 6 days and the mice body weight were recorded every day. All mice sacrificed on day 7, collected the mice blood for serum biochemical analysis and kidneys for H&E analysis, respectively.

**Statistical analysis**. Statistical analysis was performed by using two-tailed Student's *t* test for two groups and one-way analysis of variance for multiple groups. *P* values > 0.05 represented non-significance (N.S.). *P* values < 0.05 represented statistically significant, *P* values: *$p < 0.05$, **$p < 0.01$, ***$p < 0.001$ (unpaired, two-tailed *t* tests).

**Reporting summary**. Further information on research design is available in the Nature Research Reporting Summary linked to this article.

## Data availability

Chemical states of elements are assigned based on the National Institute of Standards and Technology (NIST) XPS Database [https://doi.org/10.18434/T4T88K].

The source data underlying Figs. 2b–i, 3c–f, 4b, c, f–h, 5a–c, e–g, 6b–e, 7a–l, 8a–i, 9b–f, h–i and Supplementary Figs. 2a–d, 3–5, 7–10, 12–14, 16, 17, 20–22, 25, 27, 28, 31, 32, 36 are available as a Source Data file. The remaining data are available within the Article, Supplementary Information or available from the authors upon request. A reporting summary for this study is available as a Supporting Information file.

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

## Acknowledgements

We acknowledge the National Key Research and Development Program of China (No. 2017YFA0205400), Natural Science Foundation of China (NSFC) (No. 81603043, 81872811, and 31872755), and the Central Fundamental Research Funds for the Central Universities.

## Author contributions

Z.H., Y.H., and A.Y. conceived the project, designed the studies and analysed the results. Z.H., Y.W., and D.Y. carried out the experiments. Z.H., Y.W., D.Y., Y.H., and A.Y. wrote the manuscript. D.Y. created Figs. 1b and 9a. J.W., Y.H., and A.Y. supervised and revised the project.

## Competing interests

All authors declare no competing interests.
