## [Peer Review File · Nature Communications]

Reviewers' comments:

Reviewer #1 (Remarks to the Author): expertise in radiotherapy and immunotherapy

Summary

The manuscript describes the construction and testing of a novel polymer that accentuates radiation-mediated radical production and increases cancer cell death in vitro. Using immunocompetent mice and syngeneic tumor models, the manuscript demonstrates improved tumor control following systemic administration of the polymer when combined with tumor radiation. Distant tumors are also controlled by the combination, associated with increased proportions of T cells in the local and distant tumors, and control of lung metastases in a spontaneously metastatic primary tumor model.

The manuscript is generally well written, though it could benefit from some slight additional editing for use of English. The figures are clear and extensive, though the IHC images are not satisfactory. These need additional higher magnification components to allow the reader to see the results the authors describe.

The therapy appears effective in vitro and in vivo, but there are significant flaws in various of the more correlative in vivo analyses, such as the T cell infiltration and the H2AX activation, and would need more information on the cells that take up the nanoparticles in vivo to understand how this agent may be working in vivo.

The figure legends do not state whether any of the experiments have been repeated, and whether the results match. This information is essential.

Major issues

It is essential that the treatment scheme is clarified for publication. The in vivo RT dose is stated as 6Gy, at randomization (treatment d0). However, in various places the manuscript suggests a second dose was given at d6. This is not very clear and is extremely important for other researchers in the field to know this information. If animals received two doses it would be most appropriate to describe in the form of cycles of treatment, or in a more standard form such as RT 6Gy x2 with fractions delivered 6d apart (or RT 6Gy x1 if it is actually one dose).

The methods used to perform IHC in vivo are missing. These need to be inserted.

The analysis of T cell infiltration is deficient.

-The flow cytometry is poor, and the CD3 and CD4 or CD8 staining looks more like autofluorescence than actual specific staining of a T cell subpopulation.

-It is notable that the supplementary data shows that there are zero T cells in the primary or secondary CT26 tumors without treatment. This is not consistent with data from others who have used these models.

-The data is represented as percentage, without identifying percentage of what parent population. Since both CD4 and CD8 T cells are increasing in proportion, this cannot be percent of CD3 T cells, but no other markers are described in the flow panel.

-The treatment appears to result in almost all splenocytes becoming CD44+CD62L- effector phenotype. It is difficult to believe that all splenocytes are tumor-reactive following treatment. This needs to be clarified.

-These data are not suitable for publication at present.

The correlation between the effect of treatment and CD4 and CD8 T cell infiltration is interesting (with caveats as discussed above), but since the drug increases cancer cell death, it would be necessary to demonstrate whether the mechanism is actually dependent on T cells in vivo. CD4 or

CD8 depletion studies would demonstrate whether primary tumor control occurs via T cell responses and identify the component of tumor control that relates to increased cancer cell death.

The analysis of RT-induced cytotoxicity with drug would be best represented as a clonogenic assay. This is the gold standard for assessing radiosensitizers and would pair well with the existing data to understand the effect of treatment on early immunogenic cell death versus overall clonogenic activity across a range of RT doses.

The H2AX IHC data are problematic. Activation of H2AX is very rapid following RT, but the figure examines p-H2AX 21 day following RT. It is not plausible that this relates to the RT-induced DNA damage, and if real must relate to secondary effects. This must be clarified.

The drug has a clear dose-dependent effect in vitro, but it is unclear what tumor dose is achieved in vivo. Since the authors have imaging information, it would be valuable to calculate what proportion of the in vitro active dose is achieved in the tumor following systemic application.

There needs to be some analysis of the cell types taking up the nanoparticle in vivo. Gadolinium is used as a comparator for uptake studies in vivo, and this is mostly taken up by phagocytic cells – in particular macrophages. It is reasonable that the agent is selectively taken up by tumor-infiltrating macrophages rather than cancer cells in the tumor in vivo. This would significantly change the interpretation of the data.

Figure 8 is cut off in the manuscript file. The bottom part cannot currently be reviewed. It appears to show liver metastases from 4T1 tumors, but would need to be provided for review.

Minor issues.

The dosing and timing used in experiments is not well communicated in the main manuscript or the figure legends. This information is present in the methods, so this is a minor issue. It would be much clearer if the figure legends communicated the RT dose given and the timing when samples were harvested for analysis.

The dose and location of 4T1 injection should be provided.

The authors should discuss in greater depth how this agent differs from similar radiosensitizers that have been applied in preclinical models

Reviewer #2 (Remarks to the Author): expertise in nanoparticles

The paper presents some very interesting pre-clinical data and the therapeutic results are really excellent.

Anyway, the biodistribution of the particles, and more precisely of the gadolinium, is not clear.

- Is the gadolinium stable in the particles ?
- Do we observe any trans-metallation after injection ?
- What is the final distribution of the gadolinium ?
- The sizes of the particles are supposed to be large (more than 100 nm), the authors said that such particles were metabolized through the kidney. It is a very large size, and it's really difficult to believe it may happen without a degradation of the particle and then some risks of "free gadolinium" in the circulation (or low chelates stability). This point is really important and should be studied.
- If it is a stability phenomenon, Hemin@Gd-NCPs and Gd-NCPs should present a different stability and degradation process, and then a lot of observed differences may just come from these points.

Response to Referees

Dear Reviewers:

Thank you for your good comments concerning our manuscript entitled “Nanoscale coordination polymers induce immunogenic cell death by amplifying radiation therapy mediated oxidative stress” (ID: NCOMMS-20-00266). These comments are all valuable and very helpful for revising and improving our paper, as well as the important guiding significance to our researches. We have studied comments carefully and have made correction which we hope meet with approval. Revised portions are marked in yellow in the paper. The point to point responses to the reviewer's comments are listed as following:

Reviewer #1 (expertise in radiotherapy and immunotherapy):

Comment 1: The manuscript is generally well written, though it could benefit from some slight additional editing for use of English. The figures are clear and extensive, though the IHC images are not satisfactory. These need additional higher magnification components to allow the reader to see the results the authors describe.

Response: We carefully polished the English expression again with the help of professional Editing Services. We have also replaced with higher magnification IHC images of CD4/CD8⁺ T cells in the Supplementary Information 21. Thanks for the reviewer's carefulness.

Comment 2: The figure legends do not state whether any of the experiments have been repeated, and whether the results match. This information is essential.

Response: We are very sorry for our negligence. We have supplemented these statements in the figure legends where the experiments were repeated. The modified figure legends included: Fig 2h, 2i; 3a-3f; 4c; 5d; 6a, 6c-6e; 7i, 7j.

Comment 3: It is essential that the treatment scheme is clarified for publication. The *in vivo* RT dose is stated as 6 Gy, at randomization (treatment d0). However, in various places the manuscript suggests a second dose was given at d6. This is not very clear and is extremely important for other researchers in the field to know this information. If animals received two doses it would be most appropriate to describe in the form of cycles of treatment, or in a more standard form such as RT 6Gy ×2 with fractions delivered 6d apart (or RT 6Gy ×1 if it is actually one dose).

**Response:** The reviewer's suggestion is really beneficial and professional to clarify
the treatment scheme. According to the reviewer's constructive advice, we have
provided the clear description about RT dose in the whole manuscript.

**Comment 4:** The methods used to perform IHC in vivo are missing. These need to be
inserted.

**Response:** Thanks for the reviewer's carefulness. We carefully re-checked the whole
manuscript, and found that the methods of IHC (Ki67, CD4, CD8) and
immunofluorescence (TUNEL, γ -H2AX, CD4, CD8) were missing. We have inserted
these staining methods in the manuscript and marked it in yellow (Page 29, 31
Methods of Manuscript).

**Comment 5**

**Comment 5-1:** The analysis of T cell infiltration is deficient. The flow cytometry is
poor, and the CD3 and CD4 or CD8 staining looks more like autofluorescence than
actual specific staining of a T cell subpopulation. It is notable that the supplementary
data shows that there are zero T cells in the primary or secondary CT26 tumors
without treatment. This is not consistent with data from others who have used these
models.

**Response:** Thanks for your valuable comments. The way of the flow cytometric data
we displayed may lead to misunderstandings by reviewers and readers. The same data
was presented differently in “Smooth” and “Pseudocolor” mode of FlowJo, but the
statistics are consistent. In “Smooth” mode, some discrete cells would be ignored,
making them invisible. Acutally, the ratios of CD8⁺ T cells in primary CT26 tumors
without treatment were 0.26, 0.67, 0.54, 0.53, 0.46, respectively. The ratios of CD8⁺ T
cells in secondary CT26 tumors without treatment were 0.02, 0.89, 0.18, 0.79, 0.85,
respectively. According to the reviewer's advice, we changed the “Smooth” pattern to
“Pseudocolor” mode in Supplementary Figure 18, 19 and Table 2, 3.

Supplementary Figure 18. CD3⁺ CD4⁺ T cells detected by Flow cytometry with different treatments in CT26-bearing mice.

Supplementary Table 2. The ratios of CD4⁺ T cells in primary and distant tumors detected by Flow cytometry with different treatments in CT26-bearing mice.

CD4 ⁺ T cells in Primary tumor (%)	1	2	3	4	5	Mean ± SEM
Saline	0.47	1.40	1.57	2.01	2.24	1.54 ± 0.31
Saline+RT	0.60	1.29	1.80	1.99	2.10	1.55 ± 0.28
αPD-L1+RT	2.28	2.49	2.76	2.82	2.90	2.65 ± 0.12
H@Gd-NCPs+RT	3.64	3.14	4.35	4.62	4.42	4.03 ± 0.28
H@Gd-NCPs+RT+αPD-L1	5.85	4.63	6.75	5.79	6.71	5.95 ± 0.39
CD4 ⁺ T cells in Distant tumor (%)	1	2	3	4	5	Mean ± SEM
Saline	0.07	0.33	1.19	1.9	2.32	1.16 ± 0.43
Saline+RT	0.13	0.33	1.15	1.5	2.12	1.05 ± 0.37
αPD-L1+RT	2.32	2.54	2.80	2.47	3.10	2.64 ± 0.14
H@Gd-NCPs+RT	3.38	3.85	4.14	4.27	4.35	3.99 ± 0.18
H@Gd-NCPs+RT+αPD-L1	6.72	6.7	5.85	5.75	4.89	5.98 ± 0.34

Supplementary Figure 19. CD8⁺ CD3⁺ T cells detected by Flow cytometry (FCM) with different treatments in CT26-bearing mice.

Supplementary Table 3. The ratios of CD8⁺ T cells in primary and distant tumors detected by Flow cytometry with different treatments in CT26-bearing mice.

CD8 ⁺ T cells in Primary tumor (%)	1	2	3	4	5	Mean ± SEM
Saline	0.26	0.67	0.54	0.53	0.46	0.49 ± 0.07
Saline+RT	0.46	0.67	0.67	0.61	0.37	0.56 ± 0.06
αPD-L1+RT	0.99	0.83	0.85	0.86	0.99	0.91 ± 0.03
H@Gd-NCPs+RT	1.01	1.54	1.46	1.21	1.24	1.29 ± 0.09
H@Gd-NCPs+RT+αPD-L1	1.91	1.91	2.05	2.14	1.70	1.94 ± 0.07
CD8 ⁺ T cells in Distant tumor (%)	1	2	3	4	5	Mean ± SEM
Saline	0.02	0.89	0.18	0.79	0.85	0.55 ± 0.19
Saline+RT	0.14	0.79	0.78	0.82	0.92	0.69 ± 0.14
αPD-L1+RT	1.02	0.97	1.04	1.06	1.10	1.04 ± 0.02
H@Gd-NCPs+RT	1.54	1.61	1.77	1.67	1.55	1.63 ± 0.04
H@Gd-NCPs+RT+αPD-L1	2.29	2.23	2.31	2.23	2.47	2.31 ± 0.04

**Comment 5-2:** The data is represented as percentage, without identifying percentage
of what parent population. Since both CD4 and CD8 T cells are increasing in
proportion, this cannot be percent of CD3 T cells, but no other markers are described
in the flow panel.

**Response:** We are very sorry for the negligence. The parent population is all the cells
harvested from tumor tissues (Supplementary Figure 18, 19). To identify CD4⁺ or
CD8⁺ T cells, we labelled these T cells with anti-CD3 antibody and anti-CD4
antibody or anti-CD8 antibody. Then, the ratio of CD3⁺CD4⁺ and CD3⁺CD8⁺ T cells
in tumor tissues was added in Supplementary Table 2 and 3.

**Comment 5-3:** The treatment appears to result in almost all splenocytes becoming
CD44⁺ CD62L⁻ effector phenotype. It is difficult to believe that all splenocytes are
tumor-reactive following treatment. This needs to be clarified.

**Response:** We are very sorry for the negligence to lead to misunderstanding. In this
experiments, the parent population is CD3⁺ CD8⁺ T cells, and CD44⁺ CD62L⁻ effector
phenotype percentage refers to the ratio of CD44⁺ CD62L⁻ effector memory T cells in
CD3⁺ CD8⁺ T cells. We added the parent population information and statistical data in
Supplementary Figure 23 and Table 4.

**Supplementary Figure 23.** CD44⁺ CD62L⁻ effector memory T cells in in CD3⁺ CD8⁺ T cells
detected by Flow cytometry in spleen of treated CT26-bearing mice.

Supplementary Table 4. The ratios of effector memory T cells in spleen detected by Flow cytometry with different treatments in CT26-bearing mice.

CD44 ⁺ CD62L ⁻ effector memory T cells in CD3 ⁺ CD8 ⁺ T cells (%)	1	2	3	4	5	6	Mean ± SEM
Saline	25.0	15.3	29.1	13.8	27.3	23.3	22.3 ± 2.6
Saline+RT	16.7	25.2	36.7	26.3	25.0	15.4	24.2 ± 3.1
αPD-L1+RT	39.1	35.1	39.6	35.6	34.4	41.2	37.5 ± 1.1
H@Gd-NCPs+RT	54.7	54.6	50.5	54.2	56.5	50.8	53.6 ± 1.0
H@Gd-NCPs+RT+αPD-L1	70.6	75.2	57.5	66.1	61.5	56.5	64.6 ± 3.0

**Comment 6:** The correlation between the effect of treatment and CD⁺4 and CD8⁺ T
cells infiltration is interesting (with caveats as discussed above), but since the drug
increases cancer cell death, it would be necessary to demonstrate whether the
mechanism is actually dependent on T cells *in vivo*. CD4 or CD8 depletion studies
would demonstrate whether primary tumor control occurs *via* T cell responses and
identify the component of tumor control that relates to increased cancer cell death.

**Response:** Thanks for the reviewer's constructive comments. According to the
reviewer's valuable suggestion and other previous reports, we speculated that CD8⁺ T
cells would play a more important role in tumor inhibition. Then we performed the
CD8⁺ T cells depletion experiment on a bilateral model of CT26 tumors.

As shown in Fig. 8, we observed that Hemin@Gd-NCPs+RT treatment lost most of
the immunotherapeutic effect in primary CT26 tumors after CD8⁺ T cells depletion.
Furthermore, in secondary tumors, CD8⁺ T cells depletion completely eliminated the
therapeutic effect of Hemin@Gd-NCPs+RT. We then analyzed infiltrating cytotoxic
CD8⁺ T cells in primary and distant tumors, respectively. Hemin@Gd-NCPs
sensitized irradiation remained the effective CD8⁺ T cell infiltration (1.26% in
primary tumors and 1.68% in distant tumors), when compared with control or
radiation therapy alone. However, αCD8a treatment also significantly eliminated
Hemin@Gd-NCPs+RT mediated CD8⁺ T cell infiltration in primary (0.21%) and
distant (0.30%) tumors, respectively. These results indicated that CD8⁺ T cells deeply
involved in Hemin@Gd-NCPs mediated radiation sensitization and
immunotherapeutics (Fig. 8a-8h, Supplementary Table 5). We have discussed the
results in the Page 18 of Manuscript and inserted the detailed methods in the Page 31
of Manuscript, respectively.

Fig. 8 CD8⁺ T cells depletion experiments and *ex vivo* analysis of immune cells. (a, b) Primary (a) and distant (b) tumor growth curves of CT26 colorectal bilateral tumor-bearing mice treated with Saline, Saline+RT, Hemin@Gd-NCPs+RT and Hemin@Gd-NCPs+RT+ α CD8a (n=8). [Hemin]=12.5 mg/kg, [Gd³⁺]=30 mg/kg and [α CD8a]=10 mg/kg. Treatments were performed on days 0 and 6. X-ray radiation therapy was performed 6 hours after nanomedicines intravenous injection (black arrow), RT 6Gy \times 2 with fractions delivered 6 days apart, only primary tumors received radiation therapy. Anti-CD8a antibody was treated *via* intraperitoneal injection 6 hours after radiation therapy (red arrow). Data (a, b) were shown as mean \pm SEM. (c, d) Primary (c) and distant (d) CT26 tumor weight (n=8). (e, f) Growth curves of primary (e) and distant (f) individual tumors in the Hemin@Gd-NCPs+RT and Hemin@Gd-NCPs+RT+ α CD8a groups. (g, h) The percentages of CD8⁺ T cells in the primary (g) and distant (h) tumors analyzed by flow cytometry (n=6). (i) The percentages of macrophages (F4/80⁺ and CD11b⁺) in the primary tumors analyzed by flow cytometry (n=6). Data (c, d, g-i) were shown as mean \pm SD. Two-sided Student's *t*-test was used to calculate statistical difference between two groups. N.S. represented nonsignificance, and **p* < 0.05, ****p* < 0.001. Source data are provided as a Source data file.

**Comment 7:** The analysis of RT-induced cytotoxicity with drug would be best
represented as a clonogenic assay. This is the gold standard for assessing
radiosensitizers and would pair well with the existing data to understand the effect of
treatment on early immunogenic cell death versus overall clonogenic activity across a
range of RT doses.

**Response:** According to the reviewer's constructive advice, we performed the cell
cloning assay to detect RT-induced long-term cytotoxicity. As shown in
Supplementary Figure 6, there were only a few viable cell colonies (12 clones) in the
H@Gd-NCPs+RT group. While in Saline, Saline+RT and Gd-NCPs+RT groups, the
tumor cell colonies were 575, 209 and 123, respectively. These results indicated that
H@Gd-NCPs could effectively sensitize radiation to prevent tumor cell proliferation.
We have discussed the results in the Page 8 of Manuscript and inserted the detailed
methods in the Page 27 of Manuscript, respectively.

**Supplementary Figure 6.** (a) Images and (b) quantification of CT26 cell clones (n=3), this
experiment was repeated twice independently with similar results and all data were shown as
mean ± SD. **p < 0.01; ***p < 0.001.

**Comment 8:** The γ -H2AX IHC data are problematic. Activation of γ -H2AX is very
rapid following RT, but the figure examines γ -H2AX 21 day following RT. It is not
plausible that this relates to the RT-induced DNA damage, and if real must relate to
secondary effects. This must be clarified.

**Response:** It was really true as the reviewer mentioned that activation of γ -H2AX
was very rapid following radiation. We are very sorry for our negligence of missing
the γ -H2AX staining methods in the manuscript, leading to the misunderstanding.
Actually, the tumor tissues used for γ -H2AX staining were harvested at 24 hours after
radiation treatment. We have inserted the detailed immunofluorescence staining
methods of γ -H2AX into the manuscript (Page 29-30 Methods of Manuscript).

**Comment 9:** The drug has a clear dose-dependent effect *in vitro*, but it is unclear
what tumor dose is achieved *in vivo*. Since the authors have imaging information, it
would be valuable to calculate what proportion of the *in vitro* active dose is achieved
in the tumor following systemic application.

**Response:** Thanks for your valuable advice. MRI information could qualitatively
determine whether there was nanomedicine in tumor tissues, but could not quantify
the drug concentration accumulated within tumor tissues.

**Supplementary Figure 7.** Pharmacokinetic study of dynamic Hemin@Gd-NCPs and Magnevist.
(a, b) UV spectrum of Gd³⁺ detection in Hemin@Gd-NCPs (a) and Magnevist (b) without burning
and nitrification. (c, d) UV spectrum of Gd³⁺ detection in Hemin@Gd-NCPs (a) and Magnevist (b)
after burning and nitrification. (e) The dynamic concentrations of Hemin@Gd-NCPs or Magnevist
accumulated in the tumor tissues. Data were shown as mean \pm SD (n=3).

Herein, we performed the drug accumulation study of Hemin@Gd-NCPs and
Magnevist in tumor tissues by a colorimetric method. After intravenous injection of
Hemin@Gd-NCPs or Magnevist, tumor tissues were respectively collected at 2, 6, 12,

24, 48 and 60 hours post administration. The concentrations of Hemin@Gd-NCPs and
Magnevist within tumor tissues were analyzed by a colorimetric method. Specifically,
Thymolphthalein Complexone (TC) was used as a colorimetric reagent to detect
gadolinium in a free state, but not in coordination state. We first tested the tumor
tissues extracts without burning and nitrification, and almost no free Gd^{3+} could be
detected in Hemin@Gd-NCPs and Magnevist groups (Supplementary Figure 7a, 7b).
While after burning and nitrification, gadolinium accumulated within tumor tissues in
both groups could be detected, respectively. The concentration of Hemin@Gd-NCPs
in the tumor tissues peaked at 6 hours ($1.04 \mu\text{mol/g}$ tumor tissue) post-injection and
maintained up to 24 hours ($0.77 \mu\text{mol/g}$ tumor tissue). While Magnevist's
concentration peaked at 2 hours ($0.74 \mu\text{mol/g}$ tumor tissue) post-injection in the tumor
regions and exhibited rapidly metabolism (Supplementary Figure 7c-7e). These
results indicated that the Hemin@Gd-NCPs and Magnevist were accumulated in the
tumor tissues in the coordination state rather than in a free state. We have discussed
the results in the Page 10 of Manuscript and inserted the detailed methods in the Page
28 of Manuscript, respectively.

**Comment 10:** There needs to be some analysis of the cell types taking up the
nanoparticle *in vivo*. Gadolinium is used as a comparator for uptake studies *in vivo*,
and this is mostly taken up by phagocytic cells-in particular macrophages. It is
reasonable that the agent is selectively taken up by tumor-infiltrating macrophages
rather than cancer cells in the tumor *in vivo*. This would significantly change the
interpretation of the data.

**Response:** We strongly agreed with the reviewer's opinion. If these
Hemin@Gd-NCPs were specifically engulfed by macrophages within the tumor
tissues and then irradiated by RT, the macrophages should be obviously damaged or
killed, which could directly affect the tumor immunological microenvironment.
Therefore, we directly detected the ratios of tumor-associated macrophages (TAMs)
within the tumor tissues after various treatments to explore this possibility. After the
treatment of RT and Hemin@Gd-NCPs+RT, their ratios of TAMs in whole tumor
tissues did not exhibit notable change, when compared with control group. These
results indicated that TAMs might not play a major part in Hemin@Gd-NCPs+RT
mediated anti-tumor effects. On the other hand, these results reminded us that the
tumor microenvironment can be modulated by targeting TAMs to further amplify the
therapeutic advantages of Hemin@Gd-NCPs in the future. These results and

discussion were added in Fig. 8i of Manuscript and Supplementary Figure 25, Table 6.

Supplementary Figure 25. F4/80⁺ and CD11b⁺ macrophages detected by Flow cytometry (FCM) with different treatments in CT26-bearing mice.

Supplementary Table 6. The ratios of F4/80⁺ and CD11b⁺ macrophages in primary and distant tumors detected by Flow cytometry with different treatments in CT26-bearing mice.

Macrophages in Primary tumors (%)	1	2	3	4	5	6	Mean ± SEM
Saline	5.17	5.59	4.96	3.62	3.20	3.04	4.26 ± 0.45
Saline+RT	5.61	4.94	4.52	4.48	3.79	3.52	4.48 ± 0.31
H@Gd-NCPs+RT	5.41	5.86	4.97	4.65	3.78	3.4	4.68 ± 0.39

Comment 10: Figure 8 is cut off in the manuscript file. The bottom part cannot currently be reviewed. It appears to show liver metastases from 4T1 tumors, but would need to be provided for review.

Response: Thanks for the reviewer's carefulness. We have reproduced typography according to the *Nature communications'* typesetting requirements. We thus provided modified Fig. 9 (Previous Fig.8) in manuscript.

Special thanks to Reviewer #1 for his/her good comments. These comments have significantly improved the quality of this paper.

**Reviewer #2 (expertise in nanoparticles):**

**Comment 1:** The dosing and timing used in experiments is not well communicated in
the main manuscript or the figure legends. This information is present in the methods,
so this is a minor issue. It would be much clearer if the figure legends communicated
the RT dose given and the timing when samples were harvested for analysis.

**Response:** Thanks for your valuable advice. We have inserted the detailed treatment
information in the figure legends (including dosing, timing and administration of
nanomedicines, RT and antibodies, respectively). The modified figure legends
included: Fig. 3, 5, 7, 8 and 9.

**Comment 2:** The dose and location of 4T1 injection should be provided.

**Response:** We are very sorry for our negligence of missing the dose and location of
4T1 injection. We have provided the information "*To evaluate the function of*
*Hemin@Gd-NCPs in inhibiting tumor metastasis, mice bearing 4T1 were established.*
*4T1 cells (5×10^5 cells each mouse) were injected subcutaneously in the right lower*
*flank of mouse.*" In the Page 32 Methods of Manuscript and marked it in yellow.
Please check it and thank you for your carefulness.

**Comment 3:** The authors should discuss in greater depth how this agent differs from
similar radiosensitizers that have been applied in preclinical models.

**Response:** We greatly agreed with the reviewer's valuable suggestion. We added the
discussion about the difference between Hemin@Gd-NCPs and other similar
radiosensitizers in Discussion section of the manuscript (Page 22-24).

Briefly, previous studied radiosensitizers (NBTXR3, AGuIX and RiMO-301) are
primarily used to sensitize radiation by depositing X-rays and their clinical benefits
are restricted to a certain extent. Besides, intratumoral administration of some
nanomedicines also severely limited their applications in different types of tumours.
Additionally, biological safety and biocompatibility were also worthy of our serious
consideration. Our established Hemin@Gd-NCPs not only took into account of the
X-ray deposition, but also had the function of GSH depletion and Magnetic
Resonance Imaging. Moreover, the synergetic therapeutic effects of
Hemin@Gd-NCPs could induce powerful ICD and potentiate checkpoint blockade
immunotherapies for systemic anti-tumor immunity.

**Comment 4**

**Comment 4-1:** Anyway, the biodistribution of the particles, and more precisely of the
gadolinium, is not clear. Is the gadolinium stable in the particles? Do we observe any
trans-metallation after injection ?

Response: We fully understood reviewer's concerns. Then, we performed the dialysis
experiments of Gd-NCPs and Hemin@Gd-NCPs to evaluate their stability. Gd-NCPs
($[Gd^{3+}] = 20 \text{ mM}$, $500 \mu\text{L}$) and Hemin@Gd-NCPs ($[Hemin] = 2 \text{ mM}$, $[Gd^{3+}] = 20 \text{ mM}$,
$500 \mu\text{L}$) were packed into dialysis bags (Solarbio, 10 kD), followed by dialysis in
50% bovine serum solution (5.0 mL) or deionized water (5.0 mL) for 7 days,
respectively. The dialysates were concentrated via vacuum distillation to 1.0 mL to
detect free Gd^{3+} by colorimetry. As shown in Supplementary Table 1, almost no free
Gd^{3+} could be detected in the dialysates after 7 days' dialysis. These results suggested
that the Gd-NCPs and Hemin@Gd-NCPs could maintain stable in deionized water or
serum. The detailed experiment method was also inserted in the "Page 25 Methods of
Manuscript".

**Supplementary Table 1.** The calculated concentration of free Gd^{3+} via UV colorimetry.

Y=0.001644X+0.02784 (SI Figure 2) $[Gd^{3+}] = (Abs_{605nm} - 0.02784) / 0.001644$	Free $[Gd^{3+}]$		
	1	2	3
Gd-NCPs in deionized water	N.D.	N.D.	N.D.
H@Gd-NCPs in deionized water	N.D.	N.D.	N.D.
Gd-NCPs in serum	N.D.	N.D.	N.D.
H@Gd-NCPs in serum	N.D.	N.D.	N.D.

*N.D.: Undetectable, below $[Gd^{3+}]$ detection limit ($6.0 \times 10^{-3} \mu\text{M}$).

To further evaluate whether the Hemin@Gd-NCPs would undergo trans-metallation
after injection, Hemin@Gd-NCPs ($[Hemin] = 2 \text{ mM}$, $[Gd^{3+}] = 20 \text{ mM}$, 1.0 mL) was
packed into dialysis bags (Solarbio, 10 kD), stirred in the 100.0 mL dialysate (50%
bovine serum, adding extra $[Na^+] = 150 \text{ mM}$, $[K^+] = 5.0 \text{ mM}$, $[Ca^{2+}] = 2.5 \text{ mM}$,
$[Mg^{2+}] = 1.25 \text{ mM}$, $[Zn^{2+}] = 30 \mu\text{M}$, $[Fe^{3+}] = 30 \mu\text{M}$, $[Cu^{2+}] = 30 \mu\text{M}$) for 7 days. The
dialysates were concentrated by vacuum distillation, and the concentrated residues
were analyzed by ICP-OES (Avio 500, USA). As shown in Supplementary Figure 4,
all the above metal ions, except Gd, could be detected by ICP-OES, which showed

that the gadolinium did not undergo obvious trans-metallation. The detailed
 experiment method and results were inserted in the “Page 26 Methods and Page 6 of
 Manuscript”.

**Supplementary Figure 4.** Trans-metallation experiments of Hemin@Gd-NCPs. (a-c) Analysis of
 metal ions content via ICP-OES.

 **Comment 4-2:** What is the final distribution of the gadolinium? The sizes of the
 particles are supposed to be large (more than 100 nm), the authors said that such
 particles were metabolized trough the kidney. It is a very large size, and it's really
 difficult to believe it may happen without a degradation of the particle and then some
 risks of "free gadolinium" in the circulation (or low chelates stability). This point is
 really important and should be study. If it is a stability phenomenon,
 Hemin@Gd-NCPs and Gd-NCPs should present a different stability and degradation
 process, and then a lot of observed differences may just come from these points.

**Response:** This allowed us to re-examine the metabolic process of the nano-drugs we
 have established. The MRI signal of Hemin@Gd-NCPs reached maximum at 6 hours
 post-injection in the tumor regions and maintained up to 24 hours (Fig. 4e, 4f of
 Manuscript). To further verify the distribution of Hemin@Gd-NCPs in the tumors, we
 also detected their accumulation *via* a colorimetric method.

**Supplementary Figure 7.** Pharmacokinetic study of dynamic Hemin@Gd-NCPs and Magnevist.

(a, b) UV spectrum of Gd³⁺ detection in Hemin@Gd-NCPs (a) and Magnevist (b) without burning

and nitrification. (c, d) UV spectrum of Gd³⁺ detection in Hemin@Gd-NCPs (a) and Magnevist (b)

after burning and nitrification. (e) The dynamic concentrations of Hemin@Gd-NCPs or Magnevist

accumulated in the tumor tissues. Data were shown as mean \pm SD (n=3).

As shown in Supplementary Figure 7, after intravenous injection of

Hemin@Gd-NCPs, tumor tissues were respectively collected from the CT26

tumor-bearing mice at 2, 6, 12, 24, 48 and 60 hours after mice sacrificed. The

concentrations of Hemin@Gd-NCPs within tumor tissues were analyzed by a

colorimetric method. The colorimetry method can only detect free Gd³⁺, but not in

coordination state. We first tested the tumor tissues without burning and nitrification,

and almost no free Gd³⁺ could be detected in Hemin@Gd-NCPs and Magnevist

groups (Supplementary Figure 7a, 7b). While after burning and nitrification,

gadolinium accumulated within tumor tissues in both groups could be detected,

respectively. The concentration of Hemin@Gd-NCPs in the tumor tissues peaked at 6

474 hours (1.04 μ mol/g tumor tissue) post-injection and maintained up to 24 hours (0.77

μ mol/g tumor tissue). While Magnevist's concentration peaked at 2 hours (0.74

μ mol/g tumor tissue) post-injection in the tumor regions and exhibited rapidly

metabolism (Supplementary Figure 7c-7e). These results indicated that the

Hemin@Gd-NCPs and Magnevist were accumulated in the tumor tissues in the

coordination state rather than in a free state. We have discussed the results in the Page

10 of Manuscript and inserted the detailed methods in the Page 28 of Manuscript,

respectively.

**Supplementary Figure 8.** (a-f) DLS data of Hemin@Gd-NCPs diluted 16, 32, 64, 128, 256 and
512 times by serum at 37 °C, respectively (n=3 biologically independent samples). (g) Histogram
of Hemin@Gd-NCPs particle size changes. (h) Line chart of Hemin@Gd-NCPs particle size
change.

Then, we tried to reveal the metabolism process of these nanomedicines via a
simulation method. Specifically, we used bovine serum albumin solution (50 mg/mL,
37 °C) as the simulated plasma to continuously dilute Hemin@Gd-NCPs. With the
process of dilution, we found that the particle size of Hemin@Gd-NCPs is gradually
decreasing from about 100 nm to 5~10 nm (512 times dilution, Supplementary Figure
8a-8h). These smaller nanoparticles could potentially be metabolized through the
kidneys.

Based on this hypothesis, we further detected the state of the metabolic products in
the urine of treated mice. We collected urine from mice at 24-48 hours after
intravenous injection of Hemin@Gd-NCPs. Similarly, we should be able to directly
detect free Gd³⁺ via the colorimetric method if there was free Gd³⁺ in the urine.
However, the results of direct testing indicated that there was almost no free Gd³⁺ in
urine (Supplementary Figure 9a).

**Supplementary Figure 9.** (a, b) UV spectrum of free [Gd³⁺] detection in urine without (a) or with
(b) burning and nitrification.

We then burned and nitrified the urine sample, which showed that there was
detectable gadolinium (Supplementary Figure 9b). Therefore, it was reasonable to
assume that Hemin@Gd-NCPs became smaller (5~10 nm) through continuous
dilution process after intravenous injection. Then, these smaller nanoparticles could
be gradually metabolized through the kidneys in the coordination state rather than in a
free state.

All these results indicated that Hemin@Gd-NCPs could maintain the coordination
state during blood circulation and even after renal excretion. We discussed this part in
the Page 10-11 of Manuscript and the detailed experiment protocols were also
inserted into the “Page 28-29 Methods of Manuscript”, please check it.

**Special thanks to Reviewer #2 for his/her good comments. These comments have**
**significantly improved the quality of this paper.**

We tried our best to improve the manuscript and made some changes in the
manuscript. These changes will not influence the content and framework of the paper.
And here we did not list the changes but marked in yellow in revised paper.

We appreciate for Reviewers' warm work earnestly, and hope that the correction
will meet with approval.

Once again, thank you very much for your comments and suggestions. These
comments have significantly improved the quality of this paper.

Best Regards

Yiqiao Hu PhD, Professor

School of Life Science and Medical School of Nanjing University, Nanjing University,
Nanjing 210093, China.

Tel: +86-25-83596143; E-mail: huyiqiao@nju.edu.cn.

REVIEWER COMMENTS

Reviewer #1 (Remarks to the Author):

Summary

The authors have been very responsive to review, clarifying the most important issues such as the treatment scheme, experimental repeats, and missing timing and methods. The manuscript remains strong and has value.

Major issues

There remain significant issues with the flow cytometry of the tumor and spleen. While there has been some clarification, the authors should know that the quality of the flow cytometry does not reach the standard needed in 2020 for publication in a major journal. The tumor flow cytometry for T cells does not adequately show distinct populations, a well-compensated background, nor sufficient markers to exclude irrelevant cells. This also applies to the spleen, which clearly shows that the voltages have not been correctly applied to identify the true CD44+CD62L- cells that are crushed on the axis. The gates in this figure are set amidst the CD44-CD62L+ naïve cells that normally make up the majority of the mouse spleen.

However, with the presence of supportive (though unquantified) tumor IHC, and the addition of the mechanistic CD8 depleting experiment, this reviewer would propose that all flow cytometry of tumors simply be deleted from the manuscript. These figures are not essential, and while the figures are in place this reviewer would say that these data should not be used to draw conclusions in any case. Therefore, to give concrete suggestions, this reviewer would propose deleting:

Supplementary Figure 15

Supplementary Figure 18

Supplementary Table 2

Supplementary Figure 19

Supplementary Table 3

Supplementary Figure 23

Supplementary Table 4

Supplementary Table 5

Supplementary Figure 25 is a different case. There are clear and discrete populations, and can more reasonably be used.

Minor issues:

Figure 6e, and description of this on p15 line 485. The western blot does not show that HMGB1 was released. Total lysates of tumors will show HMGB1 levels, but cannot distinguish whether it is inside or outside cells. Minor change to clarify that.

p15 line 510. There are no survival curves in the CT26 tumor work, and while some tumors have completely regressed at the d21 post-treatment harvest, without follow-up it is not possible to assign the animals as 'cured'. Minor change to 'tumor free at d21' or similar.

Reviewer #3 (Remarks to the Author): (to replace Reviewer #2)

1. The authors provided additional data showing the stability of the Hemin-Gd-NCP and no transmetalation occurring both in vitro and in vivo. However, based on the methodology described, mixture of GdCl₃ and 5-GMP forms precipitates, which is likely due to binding of Gd³⁺ with PO₄³⁻. However, this kind of complex is typically stable at a neutral pH, but dissociates to release free

Gd³⁺ at an acidic pH, for example, in the tumor microenvironment or intracellular endosome-lysosomal environment. Release of free Gd³⁺ has been a big concern for the Gd based contrast agent. However, there is a lack of stability study at a low pH. Moreover, even at a neutral pH, with endogenous metals such as Cu²⁺, Zn²⁺, theoretically, transmetalation can occur, specifically for this type of acyclic, less stable Gd³⁺ chelates, to release Gd³⁺, which has been well documented in the literature and release of Gd³⁺ most likely accounts for Gd based contrast agents-induced nephrogenic systemic fibrosis in clinic. It is also unclear how the Gd-GMP complex coordinates with hemin and if the iron remains within the ring and interacting with Cl⁻ after the complex formation. The stability of this Gd nanoparticle is a serious concern, which has been raised previously, but still lacks of clarity in this revision.

2. The intratumoral biodistribution and the fate of the Gd nanoparticles are still vague. It seems based on the scheme in Fig. 1 that the nanoparticles are internalized in tumor cells. What mechanisms for tumor cells not stromal cells such as macrophages to take up the nanoparticles? It would be interesting to see the cellular uptake by co-culturing tumor cells with macrophages and dosed with the Gd nanoparticle in vitro. In vivo data of intratumoral biodistribution are also lacking. Immunofluorescence staining of co-localization of the nanoparticle with tumor cells not stromal cells will be helpful.

3. The authors showed the nanoparticles likely entering the lysosome in Fig. 3. Related to the previous question, are they still stable at the extreme acidic environment in lysosome?

4. The authors had some discussion regarding how this agent differs from similar radiosensitizers such as AGuIX, indicating that the hemin is endogenous and used as a therapeutic agent, thus the Hemin Gd nanoparticles are biocompatible and biologically safer. This conclusion is not correct because the safety of this agent is highly related to the stability of Gd³⁺ in the complex.

5. From the MR images in Fig. 4, there seems extensive signal enhancement in abdominal organs at 24h up to 48h, which may suggest the catabolism of the agents in digestive organs, but surprisingly, there was no signal increase in liver. The biodistribution and metabolism of this agent remain unclear.

6. The authors clarified the irradiation dosing and schedule. The RT schedule with 2 doses of 6 Gy delivered 6 days apart does not seem a clinically relevant dose schedule. Any rationale for it?

7. The new data in Suppl Fig. 6 presented the cytotoxicity of the agent with RT in CT26 cancer cells. What was the RT dose? Similar studies with macrophages will be helpful to support the in vivo observations showing the treatment had no effect on TAM.

8. It is not clear if the flow data in Supple Fig. 18 were after fully eliminating the dead cells. Provisions of more detailed gating strategy and methodology of flow cytometry are necessary.

Response to Referees

Dear Reviewers:

Thanks a lot for your constructive comments to our manuscript entitled “Nanoscale coordination polymers induce immunogenic cell death by amplifying radiation therapy mediated oxidative stress” (ID: NCOMMS-20-00266A). These comments are very valuable and helpful for us to revise and improve the manuscript. Revised manuscript are marked in yellow in the Manuscript and Supplementary Information, and the point to point response to your comments are listed as following:

Reviewer #1 (expertise in radiotherapy and immunotherapy):

Comment 1: Therefore, to give concrete suggestions, this reviewer would propose deleting: Supplementary Figure 15 / Supplementary Figure 18 / Supplementary Table 2 / Supplementary Figure 19 / Supplementary Table 3 / Supplementary Figure 23 / Supplementary Table 4 / Supplementary Table 5.

Supplementary Figure 25 is a different case. There are clear and discrete populations, and can more reasonably be used.

Response: Thanks a lot for your preciseness and carefulness. According to your and editor's constructive suggestions, we removed the flow cytometry raw data (Supplementary Figure 15 / Supplementary Figure 18 / Supplementary Table 2 / Supplementary Figure 19 / Supplementary Table 3 / Supplementary Figure 23 / Supplementary Table 4 / Supplementary Table 5) from Supplementary Information, and provided the quantification data of IHC as **New Supplementary Figure 20 (P11, Line 351-366 of Supplementary Information, marked in yellow).**

**Supplementary Figure 20.** Quantification of CD4⁺ T and CD8⁺ T cells infiltrated in tumor tissues
based on IHC from Supplementary Figure 19. All data were shown as mean±SD. **p* < 0.05; ***p* <
0.01.

**Comment 2:** Figure 6e, and description of this on p15 line 485. The western blot does
not show that HMGB1 was released. Total lysates of tumors will show HMGB1 levels,
but cannot distinguish whether it is inside or outside cells. Minor change to clarify
that.

**Response:** Thanks for your carefulness. We extracted total protein from tumor tissue
suspensions by Cytoplasmic Protein Extraction kit (Beyotime Biotech, China). This
kit used cytoplasmic protein extraction reagents to fully swell the cells *via* low
osmotic pressure, destroy the cell membrane, release cytoplasmic proteins, and then
remove the nuclear precipitate by centrifugation. Then we clarified that in the
Manuscript as “western blot analysis of CT26 tumor tissues showed that the
extracellular and cytoplasmic HMGB1...”. (P15, Line 490-491 of Manuscript, marked
in yellow), and ‘Western Blot of HMGB1’ was added to the Methods (P32, Line
1064-1071 of Manuscript).

**Comment 3:** p15 line 510. There are no survival curves in the CT26 tumor work, and
while some tumors have completely regressed at the d21 post-treatment harvest,
without follow-up it is not possible to assign the animals as ‘cured’. Minor change to
‘tumor free at d21’ or similar.

**Response:** Thanks a lot for your preciseness. We have changed the expressions of
‘cured’ as ‘tumor free at day 21’ in the Manuscript. (P15, line 516-517 of Manuscript).

**Special thanks to Reviewer #1 for his/her good comments. These comments have**
**significantly improved the quality of this paper.**

**Reviewer #3 (expertise in nanoparticles):**

**Comment 1:** ① The authors provided additional data showing the stability of the
Hemin-Gd-NCP and no transmetalation occurring both in vitro and in vivo. However,
based on the methodology described, mixture of GdCl₃ and 5-GMP forms precipitates,
which is likely due to binding of Gd³⁺ with PO₄³⁻. However, this kind of complex is
typically stable at a neutral pH, but dissociates to release free Gd³⁺ at an acidic pH, for

example, in the tumor microenvironment or intracellular endosome-lysosomal
environment. Release of free Gd^{3+} has been a big concern for the Gd based contrast
agent. However, there is a lack of stability study at a low pH.

② Moreover, even at a neutral pH, with endogenous metals such as Cu^{2+} , Zn^{2+} ,
theoretically, transmetalation can occur, specifically for this type of acyclic, less stable
Gd^{3+} chelates, to release Gd^{3+} , which has been well documented in the literature and
release of Gd^{3+} most likely accounts for Gd based contrast agents-induced
nephrogenic systemic fibrosis in clinic.

③ It is also unclear how the Gd-GMP complex coordinates with hemin and if the iron
remains within the ring and interacting with Cl^- after the complex formation. The
stability of this Gd nanoparticle is a serious concern, which has been raised previously,
but still lacks of clarity in this revision.

① **Response:** Thanks very much for your constructive comments, which made us
realize that we should also consider the stability of Gd-NCPs and Hemin@Gd-NCPs
at an acidic pH. Then, we adjusted the pH of $GdCl_3$, Gd-NCPs and Hemin@Gd-NCPs
solutions to 7.4, 6.5, 5.0, 4.0, 3.0, 2.0, respectively, incubated for 7 days, and then
added with thymolphthalein complexon (TC) to detect free Gd^{3+} . Gd-NCPs and
Hemin@Gd-NCPs maintained the coordination state at neutral and weak acidic
($pH > 4.0$), while Gd^{3+} in $GdCl_3$ solution could be easily detected at $pH 2.0 \sim 7.4$. When
the pH value was further adjusted to below 3.0, Gd^{3+} could gradually release from
Gd-NCPs and Hemin@Gd-NCPs for TC detection (Supplementary Figure 29). These
results potentially indicated that Gd-NCPs and Hemin@Gd-NCPs could maintain the
coordination state in the blood circulation, tumor microenvironment and cell
lysosomes. We speculated that the coordination state of Gd in Gd-NCPs and
Hemin@Gd-NCPs was highly related to the pK_a of 5'-GMP ($pK_{a1} = 2.4$).
Theoretically, when $pH > 7.0$, Gd-NCPs or Hemin@Gd-NCPs maintained a relatively
stable particulate state. As the pH value gradually decreased, the phosphate was
partially mono-protonated ($4.0 < pH < 6.0$), these nanoparticles would still maintain
their particulate or coordination state. When $pH < 3.0$, free Gd^{3+} could be gradually
released from Gd-NCPs or Hemin@Gd-NCPs because of the further protonation of
phosphate groups (Supplementary Figure 30). These results were inserted into the
"Discussion" part (P24, Line 793-810 of Manuscript).

**Supplementary Figure 29.** Photographs of free Gd^{3+} detection by Thymolphthalein Complexon
(TC) under different pH values.

**Supplementary Figure 30.** Potential mechanism of the pH dependent degradation process of
Gd-NCPs or Hemin@Gd-NCPs.

② To further evaluate whether the Hemin@Gd-NCPs would undergo
trans-metallation in physiological conditions, Hemin@Gd-NCPs were packed into

dialysis bags, stirred in 100.0 mL dialysates (50% bovine serum, adding extra
 $[\text{Na}^+]=150$ mM, $[\text{K}^+]=5.0$ mM, $[\text{Ca}^{2+}]=2.5$ mM, $[\text{Mg}^{2+}]=1.25$ mM, $[\text{Zn}^{2+}]=30$ μM ,
 $[\text{Fe}^{3+}]=30$ μM , $[\text{Cu}^{2+}]=30$ μM to mimic physiological environment) for 7 days at
 pH=7.4, 6.5 and 5.0, respectively. The dialysates were collected and concentrated by
 vacuum distillation, and then the concentrates were analyzed by ICP-OES (Avio 500,
 USA). As shown in Supplementary Figure 5, all the above metal ions, except Gd^{3+} ,
 could be detected in dialysates at various pH, potentially indicating that obvious
 transmetalation process could not be proven under these applied conditions. These
 results were inserted into P6, Line 189-198 of Manuscript.

**Supplementary Figure 5.** Trans-metallation experiments of Hemin@Gd-NCPs at pH=7.4, 6.5,
 5.0, respectively. Analysis of metal ion content in trans-metallation dialysates (50% bovine serum,
 adding extra $[\text{Na}^+]=150$ mM, $[\text{K}^+]=5.0$ mM, $[\text{Ca}^{2+}]=2.5$ mM, $[\text{Mg}^{2+}]=1.25$ mM, $[\text{Zn}^{2+}]=30$ μM ,
 $[\text{Fe}^{3+}]=30$ μM , $[\text{Cu}^{2+}]=30$ μM to mimic physiological environment) *via* ICP-OES.

We fully understood the reviewer's concerns upon the stability and biosafety of
 Hemin@Gd-NCPs *in vivo*. Through our *in vitro* simulation studies, we speculated that
 Hemin@Gd-NCPs would gradually disintegrate into particulate or coordination state,
 but not free state, after intravenous administration. If a large amount of Gd^{3+} was
 released, it might cause obvious damages to normal tissues including kidneys.

Therefore, we further evaluated the acute toxicity of Hemin@Gd-NCPs and GdCl₃ in
 healthy Balb/c mice. The mice were randomly divided into three groups (n=3),
 including Saline, GdCl₃ ([Gd³⁺]=3.0 mg kg⁻¹ × 6) and Hemin@Gd-NCPs ([Gd³⁺]=30.0
 174 mg kg⁻¹ × 6, 10 times dose of GdCl₃). The mice were intravenously injected every day
 for 6 days and sacrificed on day 7, respectively. As shown in Supplementary Figure
 31, mice in GdCl₃ group exhibited obvious weight loss, while those in Saline and
 Hemin@Gd-NCPs groups did not. Serum biochemistry analysis indicated that
 renal function of the mice in GdCl₃ group were probably impaired, but no significant
 difference appeared between Saline and Hemin@Gd-NCPs groups (Supplementary
 Figure 32). Histological changes of kidneys merely occurred in GdCl₃ group, including
 multifocal chronic inflammation and interstitial edema (Supplementary Figure 33),
 which potentially indicated the relatively bio-safety of Hemin@Gd-NCPs. This
 content was inserted into the “Discussion” part (P24, Line 810-826 of Manuscript).

**Supplementary Figure 31.** Body weight change curves of individual mouse after different
 treatments.

**Supplementary Figure 32.** Serum biochemical parameters CT26-bearing mice (n=3) treated with
 Saline, GdCl₃ and Hemin@Gd-NCPs. All data were shown as mean±SD. **p < 0.01.

**Supplementary Figure 33.** H&E stain sections of kidneys treated with Saline, GdCl₃ and
Hemin@Gd-NCPs. Scale bar=50 μm.

③ Thanks for your constructive suggestions, which enabled us to investigate the
encapsulated mechanism of Hemin@Gd-NCPs. In previous study, Prof. Qu and
co-workers exhibited the schematic illustration of coordination polymer nanoparticles
formation through the self-assembly of 5'-GMP and lanthanide ions, such as Eu³⁺.
N-methylmesoporphyrin IX (NMM) was confined by π - π stacking in the nanoscale
adaptive supramolecular networks (Scheme 1)¹. Hemin (Iron protoporphyrin IX) and
NMM (N-methylmesoporphyrin IX) exhibited very similar structures and properties.
Therefore, we speculated that our established Hemin@Gd-NCPs would exhibit a
similar structure with NMM@Eu³⁺/5'-GMP, and Hemin was probably encapsulated in
the large ring formed by Gd³⁺ and 5'-GMP *via* π - π stacking. We therefore updated the
new Fig. 1a (P3, Line 79-95). This content was inserted into the “Discussion” part
(P23, Line 761-769 of Manuscript).

**Scheme 1.** Schematic illustration of coordination polymer nanoparticles formation through the
self-assembly of GMP and lanthanide ions. NMM was confined in the adaptive supramolecular
networks and showed intense luminescence. The properties were used to construct versatile logic
gates. From *Adv. Mater.* **26**, 1111-1117 (2014).

[1] Pu, F., et al. Multiconfigurable Logic Gates Based on Fluorescence Switching in Adaptive
Coordination Polymer Nanoparticles. *Adv. Mater.* **26**, 1111-1117 (2014).

**Fig. 1** (a) Schematic illustration of preparation of nanoscale coordination polymers
Hemin@Gd-NCPs.

We further detected the existence of iron and chlorine after the complex
formation by the X-ray photoelectron spectroscopy (XPS). As shown in
Supplementary Figure 4, metal element Gd with characteristic binding energy at
148.00 eV (Gd 4d_{3/2}) and Fe with characteristic binding energy at 711.75 eV (Fe 2p_{3/2}),
were consistent with standard XPS spectrum of Gd³⁺ and Fe³⁺ (NIST XPS Database).
Other non-metallic elements such as C, N, O, Cl could also be detected in Hemin or
Hemin@Gd-NCPs (Fig. 2f and Supplementary Figure 4). These results demonstrated
that Hemin molecules could remain their integrity during the complex formation. This
content was inserted into P6, Line 174-178 of Manuscript.

**Supplementary Figure 4.** Qualitative element analysis of Hemin by X-ray photoelectron
spectroscopy (XPS).

**Comment 2:** The intratumoral biodistribution and the fate of the Gd nanoparticles are
still vague. It seems based on the scheme in Fig. 1 that the nanoparticles are
internalized in tumor cells. What mechanisms for tumor cells not stromal cells such as
macrophages to take up the nanoparticles? It would be interesting to see the cellular

uptake by co-culturing tumor cells with macrophages and dosed with the Gd
nanoparticle *in vitro*. *In vivo* data of intratumoral biodistribution are also lacking.
Immunofluorescence staining of co-localization of the nanoparticle with tumor cells
not stromal cells will be helpful.

**Response:** We are very sorry to confuse the reviewer. We provided the schematic
diagram as Fig. 1 to exhibit the internalization process of Hemin@Gd-NCPs by tumor
cells to induce immunogenic cell death during radiation therapy. Actually, most of the
cells (*e.g.* tumor cells, macrophages, etc) within tumor tissues could uptake these
nanoparticles, hence we did not mention that Hemin@Gd-NCPs would be specifically
internalized by tumor cells in the manuscript. We deeply believed that the reviewer's
question was very interesting, so we further compared the internalization efficiency of
Hemin@Gd-NCPs between tumor cells and macrophages. We co-cultured tumor cells
and macrophages, dosed with Hemin@Gd-NCPs (Red) for 6 hours, and then labelled
macrophages with PE-F4/80-antibody (Yellow). As shown in Supplementary Figure
24, CT26 tumor cells exhibited obviously stronger red punctate fluorescence signals,
potentially indicating their higher internalization efficiency than macrophages
(RAW264.7 cells). This content was inserted into P20, Line 667-678 of Manuscript
and marked in yellow.

MR imaging (Fig. 4) of Hemin@Gd-NCPs *in vivo* demonstrated their intratumoral
biodistribution, and the dynamic concentrations of Hemin@Gd-NCPs detected in the
tumor tissues (Supplementary Figure 8) also qualitatively confirmed their
accumulation.

**Supplementary Figure 24.** Confocal laser scanning microscope (CLSM) images of co-cultured
CT26 and RAW264.7 cells after treatment with PE-F4/80⁺ and Hemin@Gd-NCPs, respectively.
Scale bar=10 μm.

**Comment 3:** The authors showed the nanoparticles likely entering the lysosome in
Fig. 3. Related to the previous question, are they still stable at the extreme acidic
environment in lysosome?

**Response:** As shown in Supplementary Figure 29 and 30, Gd-NCPs and
Hemin@Gd-NCPs could maintain the particulate or coordination state at pH>4.0, and
release free Gd³⁺ at pH<3.0. Therefore, we believed that these nanoparticles would
present in the particulate or coordination state, but not in free state, when located
within acidic lysosomes at pH 5.0~6.0.

**Comment 4:** The authors had some discussion regarding how this agent differs from
similar radiosensitizers such as AGuIX, indicating that the hemin is endogenous and
used as a therapeutic agent, thus the Hemin Gd nanoparticles are biocompatible and
biologically safer. This conclusion is not correct because the safety of this agent is
highly related to the stability of Gd³⁺ in the complex.

**Response:** Thanks a lot for your reminder. Hemin (PANHEMATIN®) was approved
by FDA for injection prescription medication to relieve repeated attacks of acute
intermittent porphyria (AIP). Hemin was supplied as lyophilized powder in free state
for reconstitution with sterile water just before infusion. These information indicated
that Hemin in free state displayed acceptable compatibility for *in vivo* administration.
Furthermore, acute toxicity study also confirmed the biological safety of
Hemin@Gd-NCPs even at higher cumulative dose ([Gd³⁺]=180 mg/kg). Based on
these theoretical analysis and experimental results, we believed that Gd-NCPs and
Hemin@Gd-NCPs exhibited acceptable biological safety and compatibility for *in vivo*
antitumor treatment.

Here, we must express our apology for confusion. In the Discussion (Previous P23
Line 749-750), we mentioned "... biological safety and biocompatibility were also
worthy of our consideration". We originally expressed that Gd-NCPs and
Hemin@Gd-NCPs potentially exhibited comparable and acceptable biological safety
to other Gd-based coordination molecules. This sentence might confuse the reviewer
and other readers, therefore we deleted this sentence from Discussion.

**Comment 5:** From the MR images in Fig. 4, there seems extensive signal
enhancement in abdominal organs at 24h up to 48h, which may suggest the
catabolism of the agents in digestive organs, but surprisingly, there was no signal

increase in liver. The biodistribution and metabolism of this agent remain unclear.

**Response:** Thanks for your constructive comments. We discussed with professional
radiologist, and obtained that gastrointestinal contents, including biological
macromolecules and gas, and the visceral fat surrounding gastrointestinal organs
would quickly realign its longitudinal magnetization with B₀, and exhibit extremely
strong MRI signal²⁻⁵. Therefore, the extensive signal enhancement in abdominal
organs from 24 h to 48 h, were not induced by Hemin@Gd-NCPs. Similar situations
also happened on tumor (2 h, 6 h), kidney (6 h, 12 h) in Magnevist group, and tumor
(2 h, 6 h) in Hemin@Gd-NCPs group, respectively.

Besides, some Gd-based coordination molecules or nanoparticles exhibited weak
uptake in the liver tissues, which has been previously reported by Roux and
co-workers⁶. This phenomenon could be attributed to that the Gd-based nanoparticles
could not be effectively phagocytosed by kuffer cells within the liver tissues.
Therefore, some studies had modified Gd-based nanocarriers with targeting ligands to
improve their phagocytic capacity. In our studies, it was also shown that macrophages
(RAW264.7 cells) were obviously weaker than tumor cells in phagocytosis of
Hemin@Gd-NCPs. Therefore, we speculated that insufficient phagocytosis of kuffer
cells upon Magnevist or Hemin@Gd-NCPs might be the potential reason of their low
accumulation within liver tissues.

[2] Mao, J., et al. Fat tissue and fat suppression. *J. Magn. Reson. Imaging*. **11**, (3) 385-93 (1993).

[3] Delfaut E. M., et al. Fat suppression in MR imaging: techniques and pitfalls. *Radiographics*.
**19**, (2) 373-82 (1999).

[4] De Kerviler E., et al. Fat suppression techniques in MRI: an update. *Biomed. Pharmacother*. **52**,
(2) 69-75 (1998).

[5] Bley, T. A., et al. Fat and water magnetic resonance imaging. *J. Magn. Reson. Imaging*. **31**,
4-18 (2010).

[6] Alric, C., et al. Gadolinium Chelate Coated Gold Nanoparticles As Contrast Agents for Both
X-ray Computed Tomography and Magnetic Resonance Imaging. *J. Am. Chem. Soc.* **130**, (18)
5908-5915 (2008).

**Comment 6:** The authors clarified the irradiation dosing and schedule. The RT
schedule with 2 doses of 6Gy delivered 6 days apart does not seem a clinically
relevant dose schedule. Any rationale for it?

**Response:** Hypofractionated radiotherapy (3~8 Gy per fraction) had comparable local

control capacity and side effects to standard fractionation, which was confirmed by a
number of clinical studies. In clinical practices, tumor patients sometimes received
hypofractionated radiotherapy to defense tumors, which had been widely used for
breast, bladder, thyroid and prostate cancer treatments⁷⁻¹¹. In our study, radiation (RT
6 Gy ×2 delivered 6 days apart) was performed to treat tumor-bearing mice. At the
same time, similar treatment patterns (RT 10 Gy ×2 delivered a week apart and RT 5
381 Gy ×2 delivered three days apart) often appeared in preclinical studies^{12,13}.

[7] Sanz, J. et al. Once-Weekly Hypofractionated Radiotherapy for Breast Cancer in Elderly
Patients: Efficacy and Tolerance in 486 Patients (*Clinical Study*). *Biomed Res Int*. 8321871 (2018).

[8] Zhao, M. et al. Weekly radiotherapy in elderly breast cancer patients: a comparison between
two hypofractionation schedules. *Clinical and Translational Oncology*. [https://doi.org/](https://doi.org/10.1007/s12094-020-02430-7)
10.1007/s12094-020-02430-7.

[9] Mallick, I., et al. A Phase I/II Study of Stereotactic Hypofractionated Once-weekly Radiation
Therapy (SHORT) for Prostate Cancer. *Clinical Oncology*. e39-e45 (2020).

[10] Dirix, P., et al. Hypofractionated palliative radiotherapy for bladder cancer. *Support Care*
*Cancer*. **24**, 181-186 (2016).

[11] Harriet, E.-H., et al. Patient-Reported Outcomes and Cosmesis After Once-Weekly
Hypofractionated Breast Irradiation in Medically Underserved Patients. *Int. J. Radiation Oncol.*
*Biol. Phys.* **107**, 934-942 (2020).

[12] Oweida, A., et al. Hypofractionated Radiotherapy Is Superior to Conventional Fractionation
in an Orthotopic Model of Anaplastic Thyroid Cancer. *Thyroid* **28** (6), 739-747 (2017).

[13] Gao S, et al. Selenium-Containing Nanoparticles Combine the NK Cells Mediated
Immunotherapy with Radiotherapy and Chemotherapy. *Adv. Mater.* **32**, 1907568 (2020).

**Comment 7:** The new data in Suppl Fig. 6 presented the cytotoxicity of the agent
with RT in CT26 cancer cells. What was the RT dose? Similar studies with
macrophages will be helpful to support the *in vivo* observations showing the treatment
had no effect on TAM.

**Response:** The dose of RT was 8 Gy, which had been added in the methods of *in vitro*
cytotoxicity and cloning experiments. According to the reviewer's constructive
suggestions, we then performed the *in vitro* cytotoxicity study upon tumor cells and
macrophages, respectively. Without radiation, Hemin@Gd-NCPs (0~100 μM of Gd³⁺)
did not exhibit obvious cytotoxicity to both CT26 tumor cells and RAW264.7 cells,
potentially indicating their great biocompatibility. Upon radiation, Hemin@Gd-NCPs

showed superior proliferation inhibition in CT26 tumor cells than RAW264.7 cells,
which should be probably attributed to their higher cellular internalization
(Supplementary Figure 24, 25). This content was inserted into P20, Line 667-678 of
Manuscript.

**Supplementary Figure 25.** The cytotoxicity of Hemin@Gd-NCPs against CT26 and RAW264.7
cells with or without radiation (8 Gy ×1), respectively ([Gd³⁺]=0, 12.5, 25, 50, 100 μM, n=3). This
experiment was repeated twice independently with similar results and all data were shown as
mean±SD.

**Comment 8:** It is not clear if the flow data in Supple Fig. 18 were after fully
eliminating the dead cells. Provisions of more detailed gating strategy and
methodology of flow cytometry are necessary.

**Response:** In this study, all of the flow cytometry experiments were adopted with the
same sample treatment method and gating strategy. After incubated with various
antibodies, cells were fixed by 4% paraformaldehyde and then analysed *via* flow
cytometry. During the the running process, Forward Scatter (FSC) and Side Scatter
(SSC) dot maps were established, the voltage was adjusted to ensure that all the
events were within the visible range of the dot maps. Then, the events with
appropriate FSC (200-600) and SSC (200-600) were gated and collected. Those
events with low FSC/low SSC and low FSC/high SSC were abandoned, which mainly
represented cell debris and air bubbles. This content was inserted into P32, Line
1073-1080 of Manuscript.

**Special thanks to Reviewer #3 for his/her good comments. These comments have**
**significantly improved the quality of this paper.**

We tried our best to improve the manuscript and made some modifications in the

manuscript. These changes will not influence the content and framework of the
manuscript. And we marked these changes in yellow in revised manuscript.

We appreciate for Reviewers' warm work earnestly, and hope that these corrections
will meet with approval.

Once again, thank you very much for your comments and suggestions. These
comments have significantly improved the quality of our manuscript.

Best Regards

Yiqiao Hu PhD, Professor

School of Life Science and Medical School of Nanjing University, Nanjing University,
Nanjing 210093, China.

Tel: +86-25-83596143; E-mail: huyiqiao@nju.edu.cn.

REVIEWER COMMENTS

Reviewer #3 (Remarks to the Author):

The authors have made significant revisions to this manuscript with additional data and extended discussion. They have extensively addressed the concerns and improved clarity. There are some remaining concerns as follows.

1). Experimental details need be provided in the figure captions or the main text for clarity, although some of them can be found in the Methods. For example, what radiation dose given in Fig. 6, and when immunological assays were conducted in Figs. 6 and 7.

2). The authors' response to MRI signals detected in the abdominal organs/tissues is vague. The signal enhancement was not seen in digestive tissues at baseline with either magnevist or Hemin Gd, but massive enhancement at later times, 24h and 48 h in the Hemin group, suggesting the enhancement was likely caused by the contrast agent, not intrinsic factors. As expected, the small molecule magnevist induced tissue contrast at earlier times, 2h and 6h. There is a lack of details about MRI sequences in the Method.

3). The data in Fig. 7 showed that both CD4+ and CD8+ T cells increased after the combination treatment. Radiation with/without immune checkpoint blockade has been reported to induce regulatory CD4+T cells or MDSC to hamper anticancer immune response. Was there any change in the population of CD4+ regulatory T cells after treatment?

Response to Referee

Dear Reviewer #3:

Thanks a lot for your constructive comments to our manuscript entitled “Nanoscale coordination polymers induce immunogenic cell death by amplifying radiation therapy mediated oxidative stress” (ID: NCOMMS-20-00266B). These comments are very valuable and helpful for us to revise and improve the manuscript. Revised manuscript are marked in yellow in the Manuscript and Supplementary Information, and the point to point response to your comments are listed as following:

Reviewer #3 (expertise in nanoparticles and radioimmunotherapy):

Comment 1: Experimental details need be provided in the figure captions or the main text for clarity, although some of them can be found in the Methods. For example, what radiation dose given in Fig. 6, and when immunological assays were conducted in Figs. 6 and 7.

Response: According to your constructive comments, we have added this detailed information in figure captions of Figs. 6, 7 and 8 or main text. All changes were marked in yellow (P14-17, and P20 of Manuscript).

Comment 2: The authors' response to MRI signals detected in the abdominal organs/tissues is vague. The signal enhancement was not seen in digestive tissues at baseline with either magnevist or Hemin Gd, but massive enhancement at later times, 24 h and 48 h in the Hemin group, suggesting the enhancement was likely caused by the contrast agent, not intrinsic factors. As expected, the small molecule magnevist induced tissue contrast at earlier times, 2 h and 6 h. There is a lack of details about MRI sequences in the Method.

Response: We appreciate the reviewer for the comments. To clarify whether the abdominal organs/tissues MRI signals came from Magnevist or Hemin@Gd-NCPs, we further retrospectively MR imaging of CT26-bearing mice without any treatment. During the MR Imagine (Fig. 1), we performed a total of 12 scans from lower to upper abdomen of the mouse, with an interval of 1mm between each scan. The detailed parameters used for T1-weighted imaging were as follows: flip angle=180, TR=500 ms, TE=15.0 ms, FOV=3×3, matrix=256×256, SI=1.0 mm 1.0 mm⁻¹, averages=3, slices=12, NEX=1 (P30, Line 1024-1026 of Manuscript). As shown in Fig. 1, the gastrointestinal tracts and their contents of untreated CT26-bearing mouse,

including biological macromolecules, gas, and the visceral fat, sequentially exhibited
obvious MR signals (Slices 4th-10th). Since the location, fat contents and
gastrointestinal contents of each mouse were possibly different, there would be some
differences in their MR signals. For instance, the mouse had not yet been injected
with drugs at 0 h (Fig. 4 in Manuscript, Liver imaging in the Hemin@Gd-NCPs
group), exhibiting obvious MR signal of the intestine. Herein, we added the detailed
MRI parameters in the Method (P30, Line 1024-1026 of Manuscript). Thanks again
for the Reviewer's comments.

**Figure 1.** Schematic illustration of the MRI methodology and the MR imaging of untreated
CT26-bearing mice under different slices.

**Comment 3:** The data in Fig. 7 showed that both CD4⁺ and CD8⁺ T cells increased
after the combination treatment. Radiation with/without immune checkpoint blockade
has been reported to induce regulatory CD4⁺ T cells or MDSC to hamper anticancer
immune response. Was there any change in the population of CD4⁺ regulatory T cells
after treatment?

**Response:** We appreciate for the reviewer's insightful and forward-looking comments.

At the beginning of our study, we envisioned the use of Hemin@Gd-NCPs to amplify
radiotherapy-mediated oxidative stress for immunogenic cell death induction and
CD8⁺ T-cell activation.¹⁻⁷ The experimental results also further demonstrated that the
depletion of CD8⁺ T cells almost completely eliminated the therapeutic effects of
Hemin@Gd-NCPs+RT in distal tumors (Fig. 8 in the Manuscript). Unexpectedly, we
found that amplified oxidative stress also improved the CD4⁺ T-cell infiltration in
tumor microenvironment (Fig. 7 in the Manuscript). In our another study
(unpublished) to amplify radiotherapy mediated oxidative stress, enhanced CD4⁺
T-cell infiltration was also observed, which indicated that this phenomenon was not
isolated or accidental.

Except for immune activation,⁸⁻¹² radiotherapy would also recruit
immunosuppressive cells, including Tregs and MDSCs, to mediate
radioresistance.¹³⁻¹⁷ Tregs usually account for ~4% and 20%-30% of CD4⁺ T cells in
normal tissues and tumor microenvironment, respectively.¹⁸⁻²⁰ High level Tregs in the
tumor microenvironment are associated with poor prognosis in many cancers, which
indicates that Tregs could suppress T_{eff} cells and their immune responses.²¹⁻²⁴

Here, we must say that the reviewer's speculation was very insightful and
forward-looking. When the enhanced infiltration of CD4⁺ T cells in bilateral tumor
model was observed, we also realized that Tregs might play a role in hindering the
immune response in tumor microenvironment. Subsequently, in the 4T1 metastatic
breast cancer model, we further synergized with the Treg-cell targeting antibody
αCTLA-4, which could also obviously extend the survival of mice treated by
Hemin@Gd-NCPs+RT (Fig. 9 in the Manuscript). Therefore, we cautiously
speculated that Hemin@Gd-NCPs mediated oxidative stress amplification might
enhance Treg-cell infiltration in the tumor microenvironment, thereby inducing
potential immunosuppression.

Many thanks again for the very meaningful comments, which pointing out the
direction of our future studies. This discussion have been added in P25-26, Line
857-865 of Manuscript. We intend to verify the dynamic change profiles of Tregs and
pharmacologically deplete Tregs during the process of amplifying oxidative stress in
the future studies for synergistic treatment. That would be another very interesting
area.

[1] Pluhar, G. E., et al. CD8⁺ T Cell-Independent Immune-Mediated Mechanisms of Anti-Tumor
Activity. *Crit. Rev. Immunol.* **35**, 153-172 (2015).

- [2] Farhood, B., et al. CD8⁺ cytotoxic T lymphocytes in cancer immunotherapy: A review. *J. Cell.*
*Physiol.* **234**, 8509-8521 (2019).
- [3] Raskov, H., et al. Cytotoxic CD8⁺ T cells in cancer and cancer immunotherapy. *Br. J. Cancer*
(2020). <https://doi.org/10.1038/s41416-020-01048-4>.
- [4] Dudley M. E., et al. Randomized selection design trial evaluating CD8⁺-enriched versus
unselected tumor-infiltrating lymphocytes for adoptive cell therapy for patients with melanoma. *J*
*Clin Oncol.* **31**, 2152-2159 (2013).
- [5] Klein-Hessling S., et al. NFATc1 controls the cytotoxicity of CD8⁺ T cells. *Nat. Commun.* **8**,
511 (2017).
- [6] Egelston C. A., et al. Human breast tumor-infiltrating CD8⁺ T cells retain polyfunctionality
despite PD-1 expression. *Nat. Commun.* **9**, 4297 (2018).
- [7] Leclerc, M., et al. Regulation of antitumour CD8 T-cell immunity and checkpoint blockade
immunotherapy by Neuropilin-1. *Nat. Commun.* **10**, 3345 (2019).
- [8] Delaney, G., et al. The role of radiotherapy in cancer treatment: estimating optimal utilization
from a review of evidence-based clinical guidelines. *Cancer* **104**, 1129-1137 (2005).
- [9] Demaria, S., et al. Radiotherapy: changing the game in immunotherapy. *Trends Cancer* **2**,
286-294 (2016).
- [10] Brooks, E. D.; Chang, J. Y. Time to abandon single-site irradiation for inducing abscopal
effects. *Nat. Rev. Clin. Oncol.* **16**, 123-135 (2019).
- [11] Barker, H., Paget, J., Khan, A.; Harrington, K. J. The tumour microenvironment after
radiotherapy: mechanisms of resistance and recurrence. *Nat. Rev. Cancer* **15**, 409-425 (2015).
- [12] Rodriguez-Ruiz1, M. E., et al. Immunological impact of cell death signaling driven by
radiation on the tumor microenvironment. *Nat. Immunol.* **21**, 120-134 (2020).
- [13] Oweida A. J., Darragh L., Phan A., et al. STAT3 modulation of regulatory T cells in response
to radiation therapy in head and neck cancer. *J. Natl. Cancer I.* **111**, 1339-1349 (2019).
- [14] Oweida A., Hararah M. K., Phan A., et al. Resistance to radiotherapy and PD-L1 blockade is
mediated by TIM-3 upregulation and regulatory T-cell infiltration. *Clin. Cancer Res.* **24**,
5368-5380 (2018).
- [15] Mondini M., Loyher P. L., Hamon P., et al. CCR2-dependent recruitment of Tregs and
monocytes following radiotherapy is associated with TNF α -mediated resistance. *Cancer Immunol.*
*Res.* **7**, 376-387 (2019).
- [16] Muroyama Y., Nirschl T. R., Kochel C. M., et al. Stereotactic radiotherapy increases
functionally suppressive regulatory T cells in the tumor microenvironment. *Cancer Immunol. Res.*
**5**, 992-1004 (2017).

- [17] Beauford S S, Kumari A, Garnett-Benson C. Ionizing radiation modulates the phenotype and
function of human CD4⁺ induced regulatory T cells. *BMC Immunol.* 21, 1-13 (2020).
- [18] Bettelli E., et al. Reciprocal developmental pathways for the generation of pathogenic
effector TH17 and regulatory T cells. *Nature* **441**, 235-8 (2006).
- [19] Gooden, M. J. et al. The prognostic influence of tumour-infiltrating lymphocytes in cancer: a
systematic review with meta-analysis. *Brit. J. Cancer.* **105**, 93-103 (2011).
- [20] Plitas, G., et al. Regulatory T Cells in Cancer. *Annu. Rev. Cancer Biol.* **4**, 459-477 (2020).
- [21] Curiel T. J. Tregs and rethinking cancer immunotherapy. *J. Clin. Invest.* **117**, 1167-74 (2007).
- [22] Borst, J., et al. CD4⁺ T cell help in cancer immunology and immunotherapy. *Nat. Rev.*
*Immunol.* **18**, 635-647 (2018).
- [23] Kennedy, R., and Celis, E. Multiple roles for CD4⁺ T cells in anti-tumor immune responses.
*Immunol. Rev.* **222**, 129-144 (2008).
- [24] Oleinika K., et al. Suppression, subversion and escape: the role of regulatory T cells in cancer
progression. *Clin. Exp. Immunol.* **171**, 36-45 (2013).

**Special thanks to Reviewer #3 for his/her good comments. These comments have**
**significantly improved the quality of this paper and pointed out the direction of**
**our future studies.**

We tried our best to improve the manuscript and made some modifications in the
manuscript. These changes will not influence the content and framework of the
manuscript. And we marked these changes in yellow in revised manuscript.

We appreciate for Reviewers' warm work earnestly, and hope that these corrections
will meet with approval.

Once again, thank you very much for your comments and suggestions. These
comments have significantly improved the quality of our manuscript.

Best Regards

Yiqiao Hu PhD, Professor

School of Life Science and Medical School of Nanjing University, Nanjing University,
Nanjing 210093, China.

Tel: +86-25-83596143; E-mail: huyiqiao@nju.edu.cn.

REVIEWERS' COMMENTS

Reviewer #3 (Remarks to the Author):

The authors have responded to previous concerns/comments with additional data and extended discussions. In my opinion, the manuscript is now appropriate for publication in Nature Communications.